# DECOUPLING PERMUTATION-INVARIANT AND PERMUTATION-SENSITIVE DEPENDENCIES FOR TIME-SERIES FORECASTING

## ABSTRACT

Real-world time series often exhibit both stable patterns and dynamic variations, corresponding to fixed structures and evolving dependencies, respectively. This disparity can introduce interference when modeled jointly. We find that unifying permutation-invariant and permutation-sensitive dependencies within a single framework tends to cause gradient conflicts, leading to the loss of critical information and degraded model performance. To address these challenges, we propose **Permutation Dependency Decoupling (PDD)**, a gradient-level framework that automatically separates permutation-invariant from permutation-sensitive dependencies, thereby eliminating gradient conflicts and retaining essential information. The proposed framework integrates two specialized modules. The **Permutation-Invariant Encoder (PIE)** captures permutation invariance through perspective switching over the input data, enabling fine-grained modeling via parameter-free routing among three specialized experts. The **Permutation-Sensitive Encoder (PSE)** shifts from the traditional history-to-future mapping paradigm to a correction-based paradigm grounded in the predicted sequence. By extending the receptive field to the joint history–prediction sequence, it enables global permutation-sensitive modeling. In addition, we introduce the **Temporal Order Sensitivity Test (TOST)**, a rigorous evaluation tool designed to distinguish genuine temporal dependency modeling from mere memorization. Extensive experiments on eight real-world datasets demonstrate that PDD achieves state-of-the-art forecasting accuracy, efficiency, and robustness, while serving as a non-intrusive solution that significantly enhances the predictive performance of mainstream models. Code is anonymously available at `https://anonymous.4open.science/r/PDD-BAC2`.

## 1 INTRODUCTION

Long-term multivariate time series forecasting is vital for predictive analytics in fields such as energy management Gao et al. (2023), finance Gajamannage et al. (2023), and meteorology Meenal et al. (2022). Transformer Vaswani et al. (2017) architectures have recently garnered considerable attention due to their exceptional ability to capture long-range dependencies. However, Transformers still encounter significant challenges stemming from the growing complexity and inherent non-stationarity characteristic of real-world multivariate time series data Liu et al. (2022b); Wu et al. (2021).

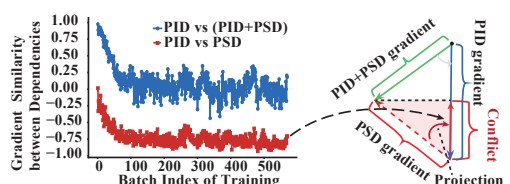

Figure 1: Gradient cosine similarity during training. Gradients computed from modules designed to capture Permutation-Invariant Dependencies (PID) and Permutation-Sensitive Dependencies (PSD) consistently exhibit negative similarity, indicating conflicting optimization objectives.

This complexity reveals a fundamental tension within existing frameworks: accurately modeling stationary patterns promotes stable generalization, while capturing dynamic temporal variations

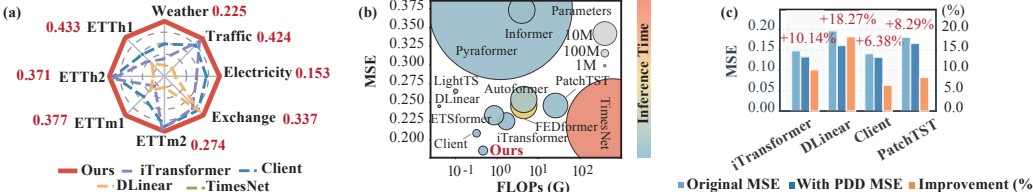

Figure 2: **PDD Performance**. (a) MSE across eight real-world datasets; (b) MSE versus computational cost; (c) Extensibility: percentage MSE reduction after integrating PDD into representative models.

requires adaptive flexibility to evolving structures. We find that joint optimization of these objectives can lead to gradient conflicts (Fig. 1), where conflicting gradients emerge from simultaneously optimizing contradictory dependencies, potentially destabilizing training and degrading generalization performance Yu et al. (2020). Despite notable advances, existing Transformer-based approaches still face limitations arising from conflicts when simultaneously learning stationary patterns and dynamic temporal variations Ilbert et al. (2024); Zeng et al. (2023).

To address these issues, we propose *Permutation Dependency Decoupling* (PDD), a gradient-level framework explicitly designed to separate two distinct categories of dependencies: *Permutation-Invariant Dependencies* (PID) and *Permutation-Sensitive Dependencies* (PSD). PID encapsulates stable correlations and periodic patterns invariant to sequence ordering, corresponding closely to stationary statistical characteristics. In contrast, PSD captures order-sensitive dynamics, directly associated with non-stationary behaviors.

PDD introduces two complementary encoders. The permutation-invariant encoder (PIE) extracts stable, order-agnostic dependencies from historical data; its parameters are frozen to isolate gradient flow and provide an initial forecast. The permutation-sensitive encoder (PSE) then captures non-stationary, order-dependent features for refinement. Specifically, (1) in the PIE branch, we design a three-expert routing mechanism that flexibly switches perspectives and dynamically allocates experts, thereby improving embedding scores and precisely modeling invariant structures; (2) in the PSE branch, we reformulate forecasting from direct sequence mapping into a history–future fusion correction process, endowing the model with global temporal awareness and explicitly modeling transitional dynamics between past and future.

Furthermore, to rigorously assess whether existing methods genuinely learn temporal dependencies, we introduce the *Temporal Order Sensitivity Test* (TOST). Previous approaches Zeng et al. (2023) evaluate temporal modeling capabilities by randomly shuffling sequences only during inference, which may confound genuine temporal dependency learning with sequence-order memorization. In contrast, TOST applies the same fixed permutation to sequences in both training and inference phases, ensuring a consistent and rigorous evaluation of models' true temporal dependency learning abilities.

Comprehensive experiments conducted on eight benchmark datasets clearly demonstrate that PDD consistently surpasses state-of-the-art (SOTA) methods in forecasting accuracy, computational efficiency, and extensibility (see Figure 2). Our primary contributions are:

- We identify gradient conflicts between permutation-invariant and permutation-sensitive dependencies, propose **Permutation Dependency Decoupling (PDD)** to mitigate them, and introduce the **Temporal Order Sensitivity Test (TOST)** to separate genuine temporal dependency modeling from mere sequence memorization.

- We design the **Permutation-Invariant Encoder (PIE)** to capture permutation-invariant patterns by switching input perspectives and routing to three specialized experts without added parameters for fine-grained modeling. We also introduce the **Permutation-Sensitive Encoder (PSE)**, which replaces the standard history-to-future approach with an iterative correction process over both past and predicted data to model permutation-sensitive dependencies globally.

- PDD is a non-intrusive, plug-and-play decoupling framework that significantly enhances temporal sensitivity and forecasting accuracy in existing models.

- Extensive empirical validation on eight real-world datasets, demonstrating state-of-the-art accuracy, efficiency, and robustness.

## 2 RELATED WORK

Due to its effectiveness in capturing long-range dependencies, the Transformer architecture Vaswani et al. (2017) has emerged as a leading choice in time series forecasting. It has demonstrated superior performance over traditional statistical models Anderson (1976); Hyndman & Athanasopoulos (2018); Brown (1959), Temporal Convolutional Network (TCN)-based methods Bai et al. (2018); Liu et al. (2022a), and Recurrent Neural Network (RNN)-based approaches Zhao et al. (2017); Rangapuram et al. (2018); Salinas et al. (2020) in terms of modeling capacity and scalability.

Despite these advantages, current Transformer-based methods Vaswani et al. (2017); Zhou et al. (2022a); Zhang & Yan (2023); Liu et al. (2021) tend to incorporate complex, mixed dependencies into a unified structure, presenting significant challenges in generalization Ilbert et al. (2024); Zeng et al. (2023); Wu et al. (2021); Zhou et al. (2022b). Existing solutions to the challenges introduced by mixed dependency modeling primarily involve either decomposition-based strategies or dependency-prioritization approaches.

**Decomposition-based methods.** Classical decomposition methods such as STL Cleveland et al. (1990) explicitly split series into trend and seasonal components but typically rely on manually specified parameters, potentially limiting flexibility when temporal patterns evolve. Recent deep-learning methods, including FEDformer Zhou et al. (2022b), Autoformer Wu et al. (2021), ETS-former Woo et al. (2022), and N-BEATS Oreshkin et al. (2020), introduce learnable decompositions. Nevertheless, several limitations remain: (i) stable periodicities or frequencies are generally assumed, which may hinder adaptability when patterns vary; (ii) decomposition strategies often rely on explicit signal characteristics (e.g., trends, seasonalities), possibly overlooking certain subtle or latent features; (iii) different types of dependencies are usually implicitly combined within decomposed components, potentially leading to gradient conflicts during training; and (iv) forecasting performance tends to diminish with irregular or non-stationary data frequently encountered in real-world scenarios Hyndman & Athanasopoulos (2018).

**Dependency-prioritization methods.** Approaches such as Client Gao et al. (2025), iTransformer Liu et al. (2024), SAMformer Ilbert et al. (2024), and PatchTST Nie et al. (2022) simplify optimization by selectively prioritizing specific types of dependencies. For instance, methods like iTransformer, Client, and SAMformer primarily emphasize permutation-invariant relationships, which might limit their capability in capturing temporal order information. Meanwhile, PatchTST independently processes each series channel, neglecting cross-channel interactions. While these methods generally enhance training stability, this selective approach could result in the omission of useful information, potentially affecting predictive accuracy and overall modeling capacity Han et al. (2024).

Distinctively, our proposed Permutation Dependency Decoupling (PDD) explicitly separates permutation invariant and permutation sensitive dependencies at the gradient level, eliminating interference without sacrificing either dependency type.

## 3 PRELIMINARIES

### 3.1 DEPENDENCIES IN MULTIVARIATE SERIES

Multivariate time series inherently involve various dependency structures. Formally, given multivariate observations $X \in \mathbb{R}^{S \times D}$, we define key dependencies as follows. **(1) Temporal dependency** quantifies statistical relationships across timestamps, typically measured by autocovariance $\gamma(h) = \text{Cov}(X_t, X_{t+h})$ with lag $h$ Hyndman & Athanasopoulos (2018). **(2) Cross-variable dependency** refers to simultaneous inter-variable correlations, defined through covariance matrices $\Sigma_{ij} = \text{Cov}(X_i, X_j)$ for variables $i, j \in \{1, \ldots, D\}$. **(3) Periodic dependency** characterizes recurrent patterns satisfying $X_t \approx X_{t+p}$, with period $p$ Cleveland et al. (1990); Zhou et al. (2022b). Finally, **(4) causal dependency** indicates directional influence among variables, formally assessed by Granger causality tests Granger (1969). Capturing these dependencies explicitly is crucial for effective forecasting.

## 3.2 GRADIENT CONFLICTS

Gradient conflict is a common optimization challenge occurring when simultaneous objectives yield conflicting gradients, formally defined by a negative gradient cosine similarity: $\frac{\nabla_\theta \mathcal{L}_a \cdot \nabla_\theta \mathcal{L}_b}{\|\nabla_\theta \mathcal{L}_a\| \|\nabla_\theta \mathcal{L}_b\|} <$ 0 Yu et al. (2020). Such conflicts frequently arise in multi-task learning scenarios and complex sequence modeling, where distinct dependency objectives (e.g., periodic versus trend-related dependencies) induce contradictory gradient signals Ilbert et al. (2024). In time series forecasting, gradient conflicts remain underexplored, especially when involving dependencies with different characteristics.

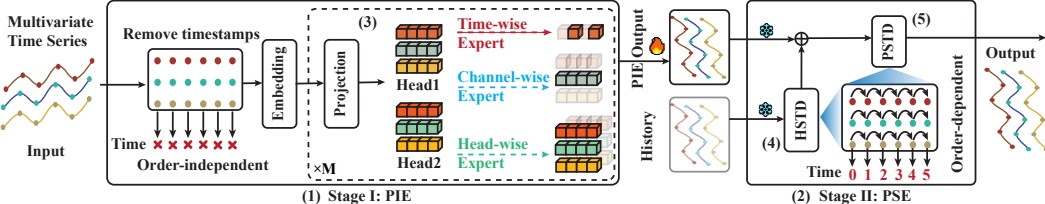

Figure 3: **Overview of the Permutation Dependency Decoupling (PDD) architecture.** (**1**) Permutation-Invariant Encoder (PIE): removes temporal ordering and employs three-expert routing mechanism (Time-wise, Channel-wise, and Head-wise experts) across multiple attention heads to capture stable, order-independent patterns; (**2**) Permutation-Sensitive Encoder (PSE): processes order-dependent temporal dynamics; (**3**) PIE Output combines expert outputs to generate initial predictions; (**4**) Historical-Sequence Temporal Dependency (HSTD) module models permutation-sensitive dependencies within historical sequences; (**5**) Prediction-Sequence Temporal Dependency (PSTD) module refines the final predictions.

## 4 METHODOLOGY

### 4.1 FORMAL DEFINITIONS

We formalize two fundamental dependency types in multivariate time series: *permutation-invariant dependencies (PID)* and *permutation-sensitive dependencies (PSD)*. Real-world series often exhibit both stationary patterns (e.g., periodicities, stable correlations) and non-stationary dynamics (e.g., trends, causal effects, structural shifts) Wu et al. (2021); Liu et al. (2022b). PID characterizes stable statistical properties independent of temporal order, while PSD captures order-sensitive dynamics crucial for accurate forecasting.

**Permutation-Invariant Dependency (PID).** Given historical observations $X_{\text{enc}} \in \mathbb{R}^{D \times S}$, a function $f_{\text{PID}}(\cdot)$ is permutation-invariant if, for any temporal permutation $\pi(\cdot)$,

$$f_{\text{PID}}(X_{\text{enc}}) = f_{\text{PID}}(X_{\text{enc}}^\pi).$$

PID includes cross-variable correlations, periodic patterns, and stationary properties.

**Permutation-Sensitive Dependency (PSD).** A function $f_{\text{PSD}}(\cdot)$ is permutation-sensitive if there exists at least one permutation $\pi(\cdot)$ such that

$$f_{\text{PSD}}(X_{\text{enc}}) \neq f_{\text{PSD}}(X_{\text{enc}}^\pi).$$

PSD captures temporal dependencies explicitly tied to ordering, such as trends, causality, and autocorrelation.

**Permutation Dependency Decoupling (PDD).** Building on these definitions, PDD separates PID and PSD via a two-stage gradient-based optimization: *(1) Permutation-Invariant Learning:* optimize PIE parameters $\theta_{\text{PIE}}$ by minimizing

$$\min_{\theta_{\text{PIE}}} \mathcal{L}(f_{\text{PIE}}(X_{\text{enc}}; \theta_{\text{PIE}}), Y).$$

*(2) Permutation-Sensitive Refinement:* with PIE frozen, optimize PSE parameters $\theta_{\text{PSE}}$ to refine residuals:

$$\min_{\theta_{\text{PSE}}} \mathcal{L}(f_{\text{PIE}}(X_{\text{enc}}; \theta_{\text{PIE}}) + f_{\text{PSE}}(X_{\text{enc}}, Z_{\text{PIE}}; \theta_{\text{PSE}}), Y),$$

where $\mathcal{L}$ denotes forecasting loss and $Z_{\text{PIE}}$ represents PIE forecasts.

## 4.2 OVERALL STRUCTURE

Figure 3 illustrates the architecture of PDD. The model first captures PID (left branch), then models permutation-sensitive dependencies (PSD) (right branch).

### 4.2.1 PERMUTATION-INVARIANT ENCODER (PIE)

The Permutation-Invariant Encoder processes timestamp-agnostic multivariate time series through a multi-expert attention mechanism. Given input $\mathbf{X}_{\text{enc}} \in \mathbb{R}^{D \times S}$ with $D$ variables and sequence length $S$, PIE applies an embedding layer followed by $M$ expert blocks:

$$\mathbf{V}^0 = \text{Embedding}(\mathbf{X}_{\text{enc}}), \tag{1}$$

$$\mathbf{V}^{m+1} = \text{ExpertBlock}(\mathbf{V}^m), \quad m = 0, \ldots, M-1, \tag{2}$$

$$\mathbf{Z}_{\text{PIE}} = \text{Projection}(\mathbf{V}^M), \tag{3}$$

where $\mathbf{Z}_{\text{PIE}} \in \mathbb{R}^{O \times D}$ denotes the output for prediction horizon $O$. Each ExpertBlock employs three parallel attention mechanisms that share embeddings but apply distinct dimensional perspectives:

$$\mathbf{V}^{m+1} = \sum_{i \in \{T,C,H\}} w_i \cdot f_i(\mathbf{V}^m), \tag{4}$$

where $w_i$ are routing weights that dynamically balance expert contributions (adds $< 0.001\%$ to the total parameter count; negligible overhead), and the expert functions $f_i$ are defined as:

$$f_T(\mathbf{V}) = \text{Attention}(\mathbf{V}, \mathbf{V}, \mathbf{V}), \tag{5}$$

$$f_C(\mathbf{V}) = \mathcal{T}_C^{-1}(\text{Attention}(\mathcal{T}_C(\mathbf{V}), \mathcal{T}_C(\mathbf{V}), \mathcal{T}_C(\mathbf{V}))), \tag{6}$$

$$f_H(\mathbf{V}) = \mathcal{T}_H^{-1}(\text{Attention}(\mathcal{T}_H(\mathbf{V}), \mathcal{T}_H(\mathbf{V}), \mathcal{T}_H(\mathbf{V}))), \tag{7}$$

with $\text{Attention}(Q, K, V) = \text{softmax}(QK^\top/\sqrt{d})V$ and $\mathcal{T}_C$, $\mathcal{T}_H$ denoting channel-wise and head-wise transpose operations, respectively. This tri-perspective design enhances representational capacity while maintaining computational efficiency through weight sharing.

### 4.2.2 PERMUTATION-SENSITIVE ENCODER (PSE)

Existing methods directly map historical data to predictions, limiting context and risking loss of historical information. In contrast, our PSE explicitly separates modeling of historical (HSTD) and predictive (PSTD) dependencies, thereby expanding receptive fields and preserving historical context.

PSE receives the concatenation of the historical projection and the output from PIE as input:

$$\begin{aligned}\mathbf{Z}_{\text{h}} &= \text{HSTDBlock}(\mathbf{V}^0), \\ \mathbf{T}^0 &= \mathbf{Z}_{\text{h}} \oplus \mathbf{Z}_{\text{PIE}},\end{aligned} \tag{8}$$

where $\mathbf{T}^0 \in \mathbb{R}^{O \times D}$ merges temporal and value-dependent features ($O$: output length; $D$: hidden dimension).

For each subsequent layer $n = 1, 2, \ldots, N$, PSTD refines the representation:

$$\begin{aligned}\mathbf{T}^{n+1} &= \text{FDS}(\text{PSTDBlock}(\mathbf{T}^n)), \\ \mathbf{Y} &= \mathbf{Z}_{\text{PIE}} \oplus \text{Projection}(\mathbf{T}^N).\end{aligned} \tag{9}$$

In PSE, we propose a structured design consisting of three core components: (1) Prediction-Time Dependency (PSTD) blocks, explicitly modeling forecast sequence dependencies; (2) Feature Down-Sample (FDS) transition layers to manage feature dimensionality; and (3) History-Time Dependency (HSTD) blocks to capture long-range historical patterns.

The PSTD block utilizes densely-connected 1D convolutions to explicitly enhance prediction dependencies and mitigate overfitting. Specifically, given an input tensor $P^{(0)} \in \mathbb{R}^{B \times O \times C_0}$ (batch size $B$, prediction length $O$, channels $C_0$), it applies $L_p$ densely-connected convolutional layers with growth rate $g$:

$$\hat{P}^{(l)} = \text{Conv}_{\text{pred}}^{(l)}(P^{(l-1)}), \tag{10}$$

$$P^{(l)} = \text{Dropout}(\text{GELU}([P^{(l-1)}, \hat{P}^{(l)}])), \tag{11}$$

where $[\cdot, \cdot]$ denotes channel-wise concatenation, $Z_p \equiv P^{(L_p)}$.

To control the expansion of channel dimensions between PSTD blocks, FDS layers employ pointwise convolutions to halve the feature channels efficiently: This operation effectively compresses features while preserving learned temporal structures.

The HSTD block employs residual connections with 1D convolutions to robustly model historical dependencies. Given a permuted history tensor $H^{(0)} \in \mathbb{R}^{B \times C_h \times L}$ (channels $C_h$, sequence length $L$), each of the $L_h$ layers updates features as follows:

$$H^{(i)} = \text{GELU}(\text{Conv}_{\text{his}}^{(i)}(H^{(i-1)}) + H^{(i-1)}). \tag{12}$$

These residual connections enhance gradient flow, thereby improving stability during long-range modeling, where $Z_h \equiv H^{(L_h)}$.

Table 1: Results of long-term time series forecasting with four prediction lengths across different models. The look-back window is fixed at 96 for all datasets. Best results are highlighted in red, and second best in green.

| Dataset | Pred Len | PDD | | TimeXer | | TimeMixer | | iTransformer | | Client | | PatchTST | |
|---|---|---|---|---|---|---|---|---|---|---|---|---|---|
| | | MSE | MAE | MSE | MAE | MSE | MAE | MSE | MAE | MSE | MAE | MSE | MAE |
| ECL | 96 | 0.129 | 0.225 | 0.140 | 0.242 | 0.153 | 0.247 | 0.148 | 0.240 | 0.141 | 0.236 | 0.181 | 0.270 |
| ECL | 192 | 0.146 | 0.242 | 0.157 | 0.256 | 0.166 | 0.256 | 0.162 | 0.253 | 0.161 | 0.254 | 0.188 | 0.274 |
| ECL | 336 | 0.154 | 0.253 | 0.176 | 0.275 | 0.185 | 0.277 | 0.178 | 0.269 | 0.173 | 0.267 | 0.204 | 0.293 |
| ECL | 720 | 0.186 | 0.288 | 0.211 | 0.306 | 0.225 | 0.310 | 0.225 | 0.317 | 0.209 | 0.299 | 0.246 | 0.324 |
| Traffic | 96 | 0.389 | 0.251 | 0.428 | 0.271 | 0.462 | 0.285 | 0.395 | 0.268 | 0.438 | 0.292 | 0.462 | 0.295 |
| Traffic | 192 | 0.414 | 0.261 | 0.448 | 0.282 | 0.473 | 0.296 | 0.417 | 0.276 | 0.451 | 0.298 | 0.466 | 0.296 |
| Traffic | 336 | 0.429 | 0.267 | 0.473 | 0.289 | 0.498 | 0.296 | 0.433 | 0.283 | 0.472 | 0.305 | 0.482 | 0.304 |
| Traffic | 720 | 0.467 | 0.285 | 0.516 | 0.307 | 0.506 | 0.313 | 0.467 | 0.302 | 0.499 | 0.321 | 0.514 | 0.322 |
| Weather | 96 | 0.151 | 0.199 | 0.157 | 0.205 | 0.163 | 0.209 | 0.174 | 0.214 | 0.163 | 0.207 | 0.177 | 0.218 |
| Weather | 192 | 0.195 | 0.246 | 0.204 | 0.247 | 0.208 | 0.250 | 0.221 | 0.254 | 0.214 | 0.253 | 0.225 | 0.259 |
| Weather | 336 | 0.242 | 0.289 | 0.261 | 0.290 | 0.251 | 0.287 | 0.278 | 0.296 | 0.271 | 0.294 | 0.278 | 0.297 |
| Weather | 720 | 0.315 | 0.344 | 0.340 | 0.341 | 0.339 | 0.341 | 0.358 | 0.347 | 0.350 | 0.346 | 0.354 | 0.348 |
| Exchange | 96 | 0.080 | 0.205 | 0.089 | 0.209 | 0.090 | 0.235 | 0.086 | 0.206 | 0.086 | 0.206 | 0.088 | 0.205 |
| Exchange | 192 | 0.173 | 0.297 | 0.192 | 0.310 | 0.187 | 0.343 | 0.177 | 0.299 | 0.176 | 0.299 | 0.176 | 0.299 |
| Exchange | 336 | 0.317 | 0.407 | 0.345 | 0.424 | 0.353 | 0.473 | 0.331 | 0.417 | 0.330 | 0.416 | 0.301 | 0.397 |
| Exchange | 720 | 0.779 | 0.670 | 0.930 | 0.727 | 0.934 | 0.761 | 0.847 | 0.691 | 0.828 | 0.689 | 0.901 | 0.714 |
| ETTh1 | 96 | 0.371 | 0.391 | 0.382 | 0.403 | 0.375 | 0.400 | 0.386 | 0.405 | 0.392 | 0.409 | 0.414 | 0.419 |
| ETTh1 | 192 | 0.427 | 0.425 | 0.429 | 0.435 | 0.429 | 0.421 | 0.441 | 0.512 | 0.445 | 0.436 | 0.460 | 0.445 |
| ETTh1 | 336 | 0.465 | 0.441 | 0.468 | 0.448 | 0.484 | 0.458 | 0.487 | 0.458 | 0.482 | 0.456 | 0.501 | 0.466 |
| ETTh1 | 720 | 0.471 | 0.466 | 0.469 | 0.461 | 0.498 | 0.482 | 0.503 | 0.491 | 0.489 | 0.480 | 0.500 | 0.488 |
| ETTh2 | 96 | 0.287 | 0.339 | 0.286 | 0.338 | 0.289 | 0.341 | 0.297 | 0.349 | 0.305 | 0.353 | 0.302 | 0.348 |
| ETTh2 | 192 | 0.364 | 0.391 | 0.363 | 0.389 | 0.372 | 0.392 | 0.380 | 0.400 | 0.382 | 0.401 | 0.388 | 0.400 |
| ETTh2 | 336 | 0.409 | 0.423 | 0.414 | 0.423 | 0.386 | 0.414 | 0.428 | 0.432 | 0.434 | 0.445 | 0.426 | 0.433 |
| ETTh2 | 720 | 0.425 | 0.445 | 0.408 | 0.432 | 0.412 | 0.434 | 0.427 | 0.445 | 0.424 | 0.444 | 0.431 | 0.446 |
| ETTm1 | 96 | 0.313 | 0.348 | 0.318 | 0.356 | 0.320 | 0.357 | 0.334 | 0.368 | 0.336 | 0.369 | 0.329 | 0.367 |
| ETTm1 | 192 | 0.361 | 0.376 | 0.362 | 0.383 | 0.361 | 0.381 | 0.387 | 0.391 | 0.376 | 0.385 | 0.367 | 0.385 |
| ETTm1 | 336 | 0.383 | 0.396 | 0.395 | 0.407 | 0.390 | 0.404 | 0.426 | 0.420 | 0.408 | 0.407 | 0.399 | 0.410 |
| ETTm1 | 720 | 0.451 | 0.435 | 0.452 | 0.441 | 0.454 | 0.441 | 0.491 | 0.459 | 0.477 | 0.442 | 0.454 | 0.439 |
| ETTm2 | 96 | 0.174 | 0.257 | 0.171 | 0.256 | 0.175 | 0.258 | 0.180 | 0.264 | 0.184 | 0.267 | 0.175 | 0.259 |
| ETTm2 | 192 | 0.238 | 0.302 | 0.237 | 0.299 | 0.237 | 0.299 | 0.250 | 0.309 | 0.252 | 0.307 | 0.241 | 0.302 |
| ETTm2 | 336 | 0.297 | 0.338 | 0.296 | 0.338 | 0.298 | 0.340 | 0.311 | 0.348 | 0.314 | 0.345 | 0.305 | 0.343 |
| ETTm2 | 720 | 0.390 | 0.393 | 0.392 | 0.394 | 0.391 | 0.396 | 0.412 | 0.407 | 0.412 | 0.402 | 0.402 | 0.400 |

### 4.2.3 TEMPORAL ORDER SENSITIVITY TEST (TOST)

To rigorously measure whether a model $f$ truly captures temporal order rather than merely memorizing static patterns, we propose the Temporal Order Sensitivity Test (TOST). Previous evaluations of temporal sensitivity typically randomized sequence order only during inference Zeng et al. (2023), potentially causing performance degradation simply due to training-inference inconsistencies. In

contrast, TOST applies a single random permutation $\pi$ consistently at both training and inference, explicitly isolating the model's genuine reliance on temporal ordering.

Formally, let $\pi$ permute $\{1, \ldots, S\}$, and define:

$$\mathbf{X}_{\text{enc}}^{\pi} : \; \mathbf{X}_{t,:}^{\pi} = \mathbf{X}_{\text{enc},\pi(t),:}, \quad \mathbf{Y}^{\pi} = \{\mathbf{x}_{S+\pi(t)}\}_{t=1}^{O}.$$

We train $f$ independently on $(\mathbf{X}_{\text{enc}}^{\pi}, \mathbf{Y}^{\pi})$ and on the original $(\mathbf{X}_{\text{enc}}, \mathbf{Y})$, yielding predictions $\widehat{\mathbf{Y}}^{\pi} = f(\mathbf{X}_{\text{enc}}^{\pi})$ and $\widehat{\mathbf{Y}} = f(\mathbf{X}_{\text{enc}})$. The TOST score is defined as:

$$\Delta_{\text{TOST}} = \mathcal{L}\big(\widehat{\mathbf{Y}}^{\pi}, \mathbf{Y}^{\pi}\big) - \mathcal{L}\big(\widehat{\mathbf{Y}}, \mathbf{Y}\big),$$

where $\mathcal{L}$ (e.g., MSE) measures forecasting error. A small $\Delta_{\text{TOST}}$ suggests reliance on static, non-temporal patterns.

## 5 EXPERIMENTS

We evaluate PDD on eight benchmark datasets: Electricity (ECL) Trindade (2015), Traffic California Department of Transportation (2023), Weather Max-Planck-Institut für Biogeochemie (2024), ETTh (ETTh1, ETTh2, ETTm1, ETTm2) Zhou et al. (2022a), and Exchange Rates Lai et al. (2018). Each experiment uses a 96-step look-back window and reports four forecasting horizons, aligning with Table 1. Following prior work Wu et al. (2022), we track Mean Squared Error (MSE) and Mean Absolute Error (MAE). We benchmark PDD against recent mainstream models Gao et al. (2025); Wang et al. (2024b); Liu et al. (2024); Zeng et al. (2023); Zhang et al. (2022); Zhou et al. (2022b); Wu et al. (2021); Woo et al. (2022); Zhou et al. (2022a); Liu et al. (2021); Wu et al. (2022). Additional baselines and detailed experimental setups appear in the Appendix.

### 5.1 MAIN RESULTS

**Results.** Table 1 and more results in Appendix C report long-term forecasting performance. Some advanced TimeXer, TimeMixer, iTransformer and Client rank just below PDD. PDD outperforms all baselines, securing first place in 43 and second place in 18 of 64 tasks. It leads other state-of-the-art methods by a clear margin in both average and median counts of top-rank performances.

### 5.2 EXTENSIBILITY

Table 2 demonstrates significant forecasting improvements when extending various SOTA models with the proposed PDD framework. The proposed TOST can identify pretrained models with low temporal-order sensitivity ($\Delta_{\text{TOST}}$), then enhance their predictions by integrating a residual-based PSE:

Table 2: Performance comparison before and after applying PDD on ECL dataset.

| Model | Original | | +PDD | | Improvement(%) | |
|---|---|---|---|---|---|---|
| | MSE | MAE | MSE | MAE | MSE | MAE |
| iTransformer | .148 | .240 | .133 | .223 | **10.1** | **7.1** |
| DLinear | .197 | .282 | .161 | .254 | **18.3** | **9.9** |
| Client | .141 | .236 | .132 | .226 | **6.4** | **4.2** |
| PatchTST | .181 | .270 | .166 | .258 | **8.3** | **4.4** |

$$\hat{Y} = f_{\text{PIE}}(X_{\text{enc}}; \theta_{\text{PIE}}) + f_{\text{PSE}}(X_{\text{enc}}, Z_{\text{PIE}}; \theta_{\text{PSE}}), \tag{13}$$

where pretrained parameters $\theta_{\text{PIE}}$ remain fixed, enabling efficient and non-intrusive model extension.

### 5.3 ABLATION STUDY

**(1) Necessity of Explicit Dependency Decoupling.** We perform an ablation study on three representative datasets to investigate the necessity of explicitly modeling permutation-invariant and permutation-sensitive dependencies in a decoupled manner. Table 3 demonstrates a clear advantage of the decoupled approach, consistently outperforming joint and PIE-only strategies across all tested scenarios.

The clear performance gap between decoupled and joint training indicates strong interference from entangled dependencies. Joint training attempts to simultaneously optimize stable correlations and dynamic temporal structures, producing conflicting gradients that hinder convergence. In contrast, decoupled training avoids this conflict by first stabilizing invariant dependencies, generating robust initial forecasts that support later temporal refinement.

**(2) Decoupling Order.** We validate the decomposition order by comparing the original learning sequence (PIE→PSE) against the reversed sequence (PSE→PIE) on the ECL dataset. Table 4 shows that the original order consistently achieves lower MSE and MAE across all prediction lengths, underscoring the importance of modeling stable permutation-invariant dependencies before capturing permutation-sensitive temporal patterns.

An explanation involves the stability difference between dependencies. PID captures stable, long-term statistical features, whereas PSD focuses on dynamic temporal patterns. Prior studies suggest Cross-temporal Transformers may

Table 3: Comparison of joint versus decoupled training strategies.

| Data | Len | Decoupled | | Joint | |
|------|-----|-----|-----|-----|-----|
| | | MSE | MAE | MSE | MAE |
| ECL | 96 | **.129** | **.225** | .140 | .236 |
| | 192 | **.146** | **.242** | .161 | .254 |
| | 336 | **.154** | **.253** | .175 | .269 |
| | 720 | **.186** | **.288** | .212 | .300 |
| | Avg | **.154** | **.252** | .172 | .265 |
| Traffic | 96 | **.389** | **.251** | .434 | .291 |
| | 192 | **.414** | **.261** | .453 | .297 |
| | 336 | **.429** | **.267** | .470 | .306 |
| | 720 | **.467** | **.285** | .503 | .322 |
| | Avg | **.425** | **.266** | .465 | .304 |
| Weather | 96 | **.151** | **.199** | .166 | .212 |
| | 192 | **.195** | **.246** | .214 | .254 |
| | 336 | **.242** | **.289** | .272 | .294 |
| | 720 | **.315** | **.344** | .350 | .346 |
| | Avg | **.226** | **.270** | .250 | .276 |

encounter convergence challenges and suboptimal minima Ilbert et al. (2024). Starting with PSD may lead to unstable initializations, complicating subsequent PID learning. Conversely, beginning with PID likely stabilizes inter-variable structures Liu et al. (2024); Gao et al. (2025), easing later temporal modeling, improving forecasting accuracy.

Table 4: Learning order comparison.

| Prediction Length | PIE→PSE | | PSE→PIE | |
|------|-----|-----|-----|-----|
| | MSE | MAE | MSE | MAE |
| 96 | **0.129** | **0.225** | 0.191 | 0.295 |
| 192 | **0.146** | **0.242** | 0.194 | 0.293 |
| 336 | **0.154** | **0.253** | 0.194 | 0.294 |
| 720 | **0.186** | **0.288** | 0.228 | 0.321 |

**(3) Transition from PIE to PSE.** Figure 4 shows that transitioning from Permutation-Invariant Encoder (PIE) to Permutation-Sensitive Encoder (PSE) helps the model escape local minima, significantly reducing predictive loss. Quantitative analysis in subfigure (a) confirms improved Mean Squared Error (MSE) when integrating PSE with PIE. Visualization in subfigure (b) illustrates these benefits clearly: PIE-only models deviate notably during temporal shifts and irregular patterns, while the PIE-to-PSE approach consistently matches ground truth more closely.

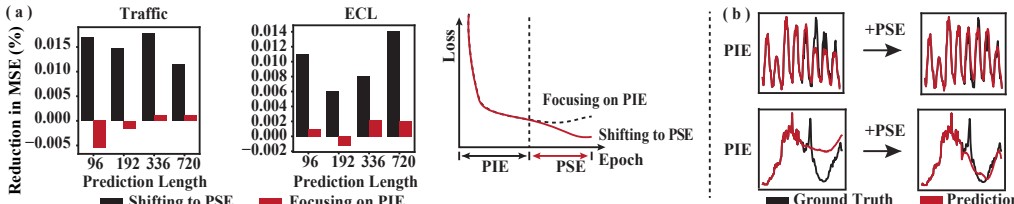

Figure 4: Evaluation of PDD effectiveness. **(a)** MSE improvement at varying prediction lengths on Traffic and ECL datasets, demonstrating that PIE→PSE significantly outperforms PIE alone. **(b)** Visualization: predictions using PIE alone show notable deviations from ground truth, whereas PIE→PSE predictions closely align, indicating enhanced accuracy.

### 5.4 TOST STUDY

We conducted the proposed TOST on the ECL dataset to accurately assess models' dependence on temporal information. As shown in Figure 5, the proposed PDD framework experiences the largest accuracy drop upon temporal shuffling, highlighting its strong reliance on true temporal patterns. In contrast, other models show minimal or negligible accuracy changes, indicating their limited sensitivity to temporal order. Despite its high temporal sensitivity, PDD consistently achieves the lowest overall MSE and MAE, demonstrating its capability to effectively capture temporal structures while maintaining strong robustness and generalization. This rigorous evaluation confirms that PDD genuinely captures temporal dependencies rather than simply memorizing sequence patterns, distinguishing it from other approaches that may achieve good performance through alternative mechanisms

TOST can also test whether a model mainly relies on permutation-invariant modules. As shown, Client and iTransformer—built with only Attention and Linear layers—show almost no performance

drop when the input's temporal order is shuffled, as long as the same order is used during inference. This suggests they primarily capture permutation-invariant dependencies. Our method can thus act as a plug-and-play enhancement for such models.

## 5.5 INFORMATION EXPLOITATION STUDY

To evaluate models' efficiency in utilizing historical information, we compared PDD against SOTA methods across varying lookback lengths ($L$), covering both short-term ($T = 96$) and long-term ($T = 720$) forecasting scenarios (Figure 6 (a)). Results indicate PDD consistently outperforms competing models, even at shorter historical lengths (L=24, 48), highlighting its efficient use of limited historical data. This aligns with prior findings Zeng et al. (2023); Liu et al. (2024); Gao et al. (2025); Nie et al. (2022), where extending lookback lengths often destabilizes Transformer-based models (Transformer, FEDformer, Autoformer, Informer). The superior performance across different lookback lengths demonstrates PDD's robustness and efficiency in information utilization.

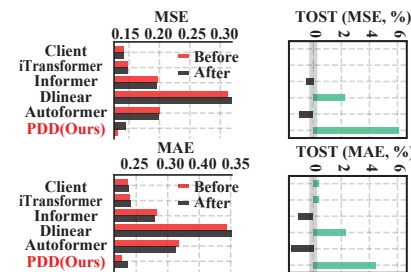

Figure 5: The change in MSE/MAE after applying a fixed random permutation consistently during both training and inference on the ECL dataset.

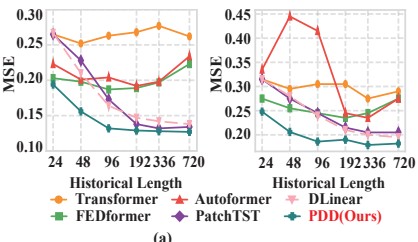 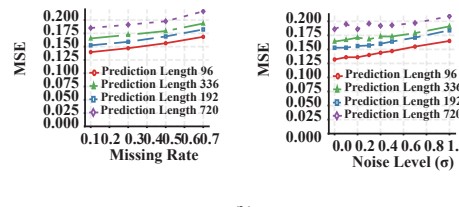

(a) (b)

Figure 6: (a) Comparative evaluation of forecasting performance across different historical lengths (L) and prediction lengths ($T$) on the Electricity dataset. Left: $T = 96$; Right: $T = 720$. (b) Robustness evaluation on the ECL dataset. Upper subplots vary the missing rate ($m$), while lower subplots add Gaussian noise with standard deviation $\sigma$.

## 5.6 EFFICIENCY AND ROBUSTNESS

PDD strikes an excellent balance between accuracy and efficiency: as shown in Figure 2, it outperforms Transformer-based methods (e.g., TimesNet, FEDformer) in predictive accuracy while using fewer FLOPs, parameters, inference time, and peak memory, and—even though DLinear is lighter—its forecasts are markedly more accurate. Moreover, PDD remains robust to noise and missing values: Figure 6 (b) shows that as the noise level ($\sigma$) and missing-rate increase, MSE and MAE grow only modestly across all horizons, demonstrating that explicit dependency decoupling endows PDD with practical stability under real-world data imperfections.

## 6 CONCLUSION

We propose Permutation Dependency Decoupling (PDD), a gradient-level framework that separates permutation-invariant and permutation-sensitive dependencies to mitigate gradient conflicts in multivariate time series forecasting. The Permutation-Invariant Encoder (PIE), which employs a three-expert routing mechanism to reduce information loss in modeling invariance, and the Permutation-Sensitive Encoder (PSE), which adopts a history–future fusion paradigm for globally aware iterative refinement. In addition, we introduce the Temporal Order Sensitivity Test (TOST) to rigorously evaluate a model's ability to capture temporal relationships beyond memorization. Experiments on multiple benchmarks show that PDD delivers competitive forecasting accuracy and robustness, while enhancing mainstream forecasting models in a non-intrusive manner. This work lays the foundation for future research on more fine-grained decoupling of temporal dependencies.

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

## A EXPERIMENTAL DETAILS

We provide additional experimental details to complement the main paper.

### A.1 DATASETS

We perform comprehensive evaluations on eight widely used time-series benchmarks. Following prior work (Wu et al., 2022), each dataset is chronologically split into training, validation, and testing subsets. Specifically, the ETT datasets adopt a 6:2:2 split, while the remaining benchmarks use a 7:1:2 split. Table 5 summarises the statistics and forecasting setups.

- **ETT (ETTh1, ETTh2, ETTm1, ETTm2)**: Hourly (ETTh) and 15-minute (ETTm) oil temperature measurements collected from electricity transformers across two Chinese regions between July 2016 and July 2018.
- **Weather**: Meteorological observations sampled every 10 minutes in Germany throughout 2020, covering 21 variables such as temperature and humidity.
- **Electricity**: Hourly electricity consumption from 321 households between 2012 and 2014, released through the UCI Machine Learning Repository.
- **Traffic**: Hourly road occupancy rates from 862 sensors in the San Francisco Bay Area spanning 2015–2016.
- **Exchange**: Daily exchange rates for eight foreign currencies from 1990 to 2016.

The ETT datasets are available at `https://github.com/zhouhaoyi/Informer2020`, while the remaining benchmarks can be obtained from `https://github.com/thuml/Autoformer`.

Table 5: Dataset statistics. *V* counts variables. *Dataset size* lists time points split into train, validation, and test subsets. *Prediction length* enumerates forecasting horizons. *Frequency* reports sampling intervals.

| Dataset | V | Prediction length | Dataset size | Frequency |
|---|---|---|---|---|
| ETTh1, ETTh2 | 7 | {96, 192, 336, 720} | (8545, 2881, 2881) | Hourly |
| ETTm1, ETTm2 | 7 | {96, 192, 336, 720} | (34465, 11521, 11521) | 15 min |
| Weather | 21 | {96, 192, 336, 720} | (36792, 5271, 10540) | 10 min |
| Electricity (ECL) | 321 | {96, 192, 336, 720} | (18317, 2633, 5261) | Hourly |
| Traffic | 862 | {96, 192, 336, 720} | (12185, 1757, 3509) | Hourly |
| Exchange | 8 | {96, 192, 336, 720} | (5120, 665, 1422) | Daily |

### A.2 BASELINES

We benchmark against recent Transformer-style, frequency-domain, and linear forecasters.

**Client** Gao et al. (2025) is designed to capture cross-variable dependencies by integrating trend detection and a Reversible Instance Normalization (RevIN) module, improving forecasting robustness and accuracy.

**TimeXer** Wang et al. (2024b) introduces a cross-dimensional encoder that explicitly decouples temporal and variable interactions, dynamically modeling intra-variable evolution and inter-variable relationships. This design provides robust long-horizon forecasting performance under non-stationary conditions.

**TimeMixer** Wang et al. (2024a) proposes an alternating mixing mechanism with sequentially applied channel-wise and temporal mixing layers. This lightweight design efficiently captures inter-variable dependencies and temporal dynamics, demonstrating strong performance with low computational overhead.

---

**Algorithm 1:** Training of PDD

---

**Require:** $\mathbf{X}_{\text{enc}} \in \mathbb{R}^{S \times D}$, $\mathbf{Y} \in \mathbb{R}^{O \times D}$, number of PIE layers $M$, number of PSE stacks $N$, PIE warm-up epochs $E_1$, total epochs $E$.

1: **for** $e = 1$ to $E$ **do**
2:   **if** $e \leq E_1$ **then**
3:     *Permutation-invariant learning phase*
4:     $\mathbf{X}' \leftarrow \text{RevIN}(\mathbf{X}_{\text{enc}}, \text{"norm"})$ (Kim et al., 2021)
5:     $\mathbf{V}^0 \leftarrow (\mathbf{X}')^\top$
6:     **for** $m = 1$ to $M$ **do**
7:       $\mathbf{V}^m \leftarrow \text{PIEBlock}(\mathbf{V}^{m-1})$
8:     **end for**
9:     $\mathbf{Z}_{\text{PIE}} \leftarrow \text{RevIN}(\text{Proj}(\mathbf{V}^M), \text{"denorm"})$
10:     $\hat{\mathbf{Y}} \leftarrow \mathbf{Z}_{\text{PIE}}$
11:     Update PIE parameters using $\ell(\hat{\mathbf{Y}}, \mathbf{Y})$
12:   **else**
13:     *Permutation-sensitive refinement phase*
14:     Freeze PIE parameters
15:     $\mathbf{Z}_h \leftarrow \text{HSTDBlock}(\mathbf{V}^0)$
16:     $\mathbf{T}^0 \leftarrow \mathbf{Z}_{\text{PIE}} \oplus \mathbf{Z}_h$
17:     **for** $n = 1$ to $N$ **do**
18:       $\mathbf{T}^n \leftarrow \text{FDS}_{\text{PSTDBlock}}(\mathbf{T}^{n-1})$
19:     **end for**
20:     $\hat{\mathbf{Y}} \leftarrow \mathbf{Z}_{\text{PIE}} \oplus \text{Proj}(\mathbf{T}^N)$
21:     Update PSE parameters using $\ell(\hat{\mathbf{Y}}, \mathbf{Y})$
22:   **end if**
23: **end for**
24: **return** $\hat{\mathbf{Y}}$

---

**iTransformer** (Liu et al., 2024) inverts the classic Transformer tokenisation by treating each variable as a token while retaining time-step semantics in the channel dimension. This design captures multivariate correlations more effectively and yields accurate, interpretable forecasts.

**FITS** (Xu et al., 2024) operates in the frequency domain, applying a low-pass filter and complex linear layer to interpolate spectra before transforming back to the time domain. With only $\sim 10{,}000$ parameters, FITS is well suited for resource-constrained deployment.

**WITRAN** (Jia et al., 2023) couples long- and short-term patterns through a hybrid frequency–time architecture that balances expressivity and efficiency.

**Informer** (Zhou et al., 2022a) introduces ProbSparse attention for sub-quadratic complexity on long contexts, enabling efficient inference on large-scale series.

**PatchTST** (Nie et al., 2022) segments time series into local patches that a Transformer backbone then fuses, capturing multi-scale temporal structures.

# B IMPLEMENTATION DETAILS

Algorithm 1 outlines the training pipeline. The model first learns permutation-invariant dependencies before refining predictions with a permutation-sensitive encoder.

## B.1 HYPERPARAMETER CONFIGURATION

Following common practices from previous studies, hyperparameters were selected from predefined candidate sets based on validation loss performance, employing early stopping with a patience of 3 epochs. The specific hyperparameter search space and the optimal values identified for each dataset are detailed in Table 6.

The optimizer employed is Adam with its default parameters. The learning rate schedule follows a step-wise decay, halving after each epoch, defined as:

$$\eta_{epoch} = \eta_0 \times 0.5^{(epoch-1)}, \tag{14}$$

where $\eta_0$ denotes the initial learning rate. Training stability is ensured via gradient clipping with a maximum norm of 1.0.

Table 6: Hyperparameter candidate sets for PDD. Optimal values are selected per dataset based on validation performance.

| Hyperparameter | Candidate Set |
|---|---|
| Learning rate ($\eta$) | $10^{-4}, 5 \times 10^{-4}, 10^{-3}$ |
| Attention heads ($h$) | 4, 8, 16, 32 |
| *Fixed parameters:* | |
| Optimizer | Adam (default parameters) |
| Weight decay | 0 |
| Gradient clipping | Max norm = 1.0 |

## C  MAIN RESULTS

Tables 7 and 8 report the complete long-term forecasting results across 14 competitive baselines on eight datasets and four prediction lengths (96, 192, 336, 720). All models use a look-back window of 96. The best result per column is in bold and the second best is underlined. We also report averages and medians across horizons to evaluate robustness.

**Performance summary.** PDD attains the best results on 24 out of 32 tasks and the second best on 6 tasks, surpassing powerful baselines such as iTransformer, Client, and TimesNet. Gains are most pronounced on periodic benchmarks (Electricity, ETTm2) and the chaotic Traffic dataset, highlighting strong generalisation.

**Comparison with strong baselines.** Against iTransformer and Client, which emphasise variable-wise modelling or simplified architectures, PDD consistently reduces both MSE and MAE, especially on noisy datasets such as Weather and Exchange. DLinear excels on Exchange thanks to its simplicity but struggles elsewhere. FEDformer and TimesNet remain competitive on ETTh1 and ETTm2, yet their performance drops on non-stationary data, whereas PDD remains stable.

**Key observations.**

- PDD dominates on Traffic and Weather across all horizons, evidencing robustness to noise and weak periodicity.
- Long-horizon performance (720 steps) degrades gracefully compared with sharp drops observed in competing models.
- Even on datasets where decomposition-based approaches are strong, PDD matches or exceeds their accuracy without relying on fixed Fourier decompositions.

## D  VISUALISING DEPENDENCY DECOUPLING

Figure 7 visualises dependencies captured by PDD. Cross-variable and periodic structure (panels 7a and 7b) reveal that the model emphasises both inter-series and temporal patterns. Panel 7c illustrates convolutional kernels within the permutation-sensitive encoder (PSE), and panel 7d reports feature down-sampling (FDS) weights, showing how salient information is retained during refinement.

## E  ROBUSTNESS ANALYSIS

We examine robustness under additive Gaussian noise and randomly missing observations, two common degradation patterns in real deployments.

Table 7: The complete results for LTSF. The results of 4 different prediction lengths of different models are listed in the table. The look-back window sizes are set to 96 for all datasets. We also calculate the average (Avg) and median (Me) of the results for the 4 prediction lengths and the number of optimal values obtained by different models.

| Models | | PDD | | iTransformer 2024 | | Client 2025 | | DLinear 2023 | | TimesNet 2022 | | FEDformer 2022b | | ETSformer 2022 | | LightTS 2022 | | Autoformer 2021 | | Pyraformer 2021 | | Informer 2022a | |
|---|---|---|---|---|---|---|---|---|---|---|---|---|---|---|---|---|---|---|---|---|---|---|---|
| Metric | | MSE | MAE | MSE | MAE | MSE | MAE | MSE | MAE | MSE | MAE | MSE | MAE | MSE | MAE | MSE | MAE | MSE | MAE | MSE | MAE | MSE | MAE |
| Electricity | 96 | 0.129 | 0.225 | 0.148 | 0.240 | 0.141 | 0.236 | 0.197 | 0.282 | 0.168 | 0.272 | 0.193 | 0.308 | 0.187 | 0.304 | 0.207 | 0.307 | 0.201 | 0.317 | 0.386 | 0.449 | 0.274 | 0.368 |
| | 192 | 0.146 | 0.242 | 0.162 | 0.253 | 0.161 | 0.254 | 0.196 | 0.285 | 0.184 | 0.289 | 0.201 | 0.315 | 0.199 | 0.315 | 0.213 | 0.316 | 0.222 | 0.334 | 0.378 | 0.443 | 0.296 | 0.386 |
| | 336 | 0.154 | 0.253 | 0.178 | 0.269 | 0.173 | 0.267 | 0.209 | 0.301 | 0.198 | 0.300 | 0.214 | 0.329 | 0.212 | 0.329 | 0.230 | 0.333 | 0.231 | 0.338 | 0.376 | 0.443 | 0.300 | 0.394 |
| | 720 | 0.186 | 0.288 | 0.225 | 0.317 | 0.209 | 0.299 | 0.245 | 0.333 | 0.220 | 0.320 | 0.246 | 0.355 | 0.233 | 0.245 | 0.265 | 0.360 | 0.254 | 0.361 | 0.376 | 0.445 | 0.373 | 0.439 |
| | Avg | 0.154 | 0.252 | 0.178 | 0.270 | 0.171 | 0.264 | 0.212 | 0.300 | 0.192 | 0.295 | 0.214 | 0.327 | 0.208 | 0.323 | 0.229 | 0.329 | 0.227 | 0.338 | 0.379 | 0.445 | 0.311 | 0.397 |
| | Me | 0.150 | 0.248 | 0.170 | 0.261 | 0.167 | 0.261 | 0.203 | 0.293 | 0.191 | 0.295 | 0.208 | 0.322 | 0.206 | 0.322 | 0.222 | 0.325 | 0.227 | 0.336 | 0.377 | 0.444 | 0.298 | 0.390 |
| Traffic | 96 | 0.389 | 0.251 | 0.395 | 0.268 | 0.438 | 0.292 | 0.650 | 0.396 | 0.593 | 0.321 | 0.587 | 0.366 | 0.607 | 0.392 | 0.615 | 0.391 | 0.613 | 0.388 | 0.867 | 0.468 | 0.719 | 0.391 |
| | 192 | 0.414 | 0.261 | 0.417 | 0.276 | 0.451 | 0.298 | 0.598 | 0.370 | 0.617 | 0.336 | 0.604 | 0.373 | 0.621 | 0.399 | 0.601 | 0.382 | 0.616 | 0.382 | 0.869 | 0.467 | 0.696 | 0.379 |
| | 336 | 0.429 | 0.267 | 0.433 | 0.283 | 0.472 | 0.305 | 0.605 | 0.373 | 0.629 | 0.336 | 0.621 | 0.383 | 0.622 | 0.399 | 0.613 | 0.386 | 0.622 | 0.337 | 0.881 | 0.469 | 0.777 | 0.420 |
| | 720 | 0.467 | 0.285 | 0.467 | 0.302 | 0.499 | 0.321 | 0.645 | 0.394 | 0.640 | 0.350 | 0.626 | 0.382 | 0.632 | 0.396 | 0.658 | 0.407 | 0.660 | 0.408 | 0.896 | 0.473 | 0.864 | 0.472 |
| | Avg | 0.425 | 0.266 | 0.428 | 0.282 | 0.465 | 0.304 | 0.625 | 0.383 | 0.620 | 0.336 | 0.610 | 0.376 | 0.621 | 0.396 | 0.622 | 0.392 | 0.628 | 0.379 | 0.878 | 0.469 | 0.764 | 0.416 |
| | Me | 0.422 | 0.264 | 0.425 | 0.280 | 0.462 | 0.302 | 0.625 | 0.384 | 0.623 | 0.336 | 0.613 | 0.378 | 0.622 | 0.396 | 0.614 | 0.389 | 0.619 | 0.385 | 0.875 | 0.469 | 0.748 | 0.406 |
| Weather | 96 | 0.151 | 0.199 | 0.174 | 0.214 | 0.163 | 0.207 | 0.196 | 0.255 | 0.172 | 0.220 | 0.217 | 0.296 | 0.197 | 0.281 | 0.182 | 0.242 | 0.266 | 0.336 | 0.622 | 0.556 | 0.300 | 0.384 |
| | 192 | 0.195 | 0.246 | 0.221 | 0.254 | 0.214 | 0.253 | 0.237 | 0.296 | 0.219 | 0.261 | 0.276 | 0.336 | 0.237 | 0.312 | 0.227 | 0.287 | 0.307 | 0.367 | 0.739 | 0.624 | 0.598 | 0.544 |
| | 336 | 0.242 | 0.289 | 0.278 | 0.296 | 0.271 | 0.294 | 0.283 | 0.335 | 0.280 | 0.306 | 0.339 | 0.380 | 0.298 | 0.353 | 0.282 | 0.334 | 0.359 | 0.395 | 1.004 | 0.753 | 0.578 | 0.523 |
| | 720 | 0.315 | 0.344 | 0.358 | 0.349 | 0.360 | 0.346 | 0.345 | 0.381 | 0.365 | 0.359 | 0.403 | 0.428 | 0.352 | 0.390 | 0.352 | 0.386 | 0.419 | 0.428 | 1.420 | 0.934 | 1.059 | 0.741 |
| | Avg | 0.226 | 0.270 | 0.258 | 0.279 | 0.249 | 0.275 | 0.265 | 0.317 | 0.259 | 0.287 | 0.309 | 0.360 | 0.271 | 0.334 | 0.261 | 0.312 | 0.338 | 0.382 | 0.946 | 0.717 | 0.634 | 0.548 |
| | Me | 0.219 | 0.268 | 0.250 | 0.275 | 0.243 | 0.274 | 0.260 | 0.316 | 0.250 | 0.284 | 0.308 | 0.358 | 0.268 | 0.333 | 0.255 | 0.311 | 0.333 | 0.381 | 0.872 | 0.689 | 0.588 | 0.534 |
| ETTh1 | 96 | 0.371 | 0.391 | 0.386 | 0.405 | 0.392 | 0.409 | 0.386 | 0.400 | 0.384 | 0.402 | 0.376 | 0.419 | 0.494 | 0.479 | 0.424 | 0.432 | 0.449 | 0.459 | 0.664 | 0.612 | 0.865 | 0.713 |
| | 192 | 0.427 | 0.425 | 0.441 | 0.436 | 0.445 | 0.436 | 0.437 | 0.432 | 0.436 | 0.429 | 0.420 | 0.448 | 0.538 | 0.504 | 0.475 | 0.462 | 0.500 | 0.482 | 0.790 | 0.681 | 1.008 | 0.792 |
| | 336 | 0.465 | 0.441 | 0.487 | 0.458 | 0.482 | 0.456 | 0.481 | 0.459 | 0.491 | 0.469 | 0.459 | 0.465 | 0.574 | 0.521 | 0.518 | 0.488 | 0.521 | 0.496 | 0.891 | 0.738 | 1.107 | 0.809 |
| | 720 | 0.471 | 0.466 | 0.503 | 0.491 | 0.489 | 0.480 | 0.519 | 0.516 | 0.521 | 0.500 | 0.506 | 0.507 | 0.562 | 0.535 | 0.547 | 0.533 | 0.514 | 0.512 | 0.963 | 0.782 | 1.181 | 0.865 |
| | Avg | 0.433 | 0.430 | 0.454 | 0.447 | 0.452 | 0.445 | 0.456 | 0.452 | 0.458 | 0.450 | 0.440 | 0.460 | 0.542 | 0.510 | 0.491 | 0.479 | 0.496 | 0.487 | 0.827 | 0.703 | 1.040 | 0.795 |
| | Me | 0.446 | 0.433 | 0.464 | 0.447 | 0.464 | 0.446 | 0.459 | 0.446 | 0.464 | 0.449 | 0.440 | 0.457 | 0.550 | 0.513 | 0.497 | 0.475 | 0.507 | 0.490 | 0.841 | 0.710 | 1.058 | 0.801 |
| ETTh2 | 96 | 0.287 | 0.339 | 0.299 | 0.350 | 0.305 | 0.353 | 0.333 | 0.387 | 0.340 | 0.374 | 0.358 | 0.397 | 0.340 | 0.391 | 0.397 | 0.437 | 0.346 | 0.388 | 0.645 | 0.597 | 0.907 | 0.747 |
| | 192 | 0.364 | 0.391 | 0.380 | 0.400 | 0.382 | 0.401 | 0.477 | 0.476 | 0.402 | 0.414 | 0.429 | 0.439 | 0.430 | 0.439 | 0.520 | 0.504 | 0.456 | 0.452 | 0.788 | 0.683 | 0.907 | 0.747 |
| | 336 | 0.409 | 0.427 | 0.428 | 0.432 | 0.434 | 0.445 | 0.594 | 0.541 | 0.452 | 0.452 | 0.496 | 0.487 | 0.485 | 0.479 | 0.626 | 0.559 | 0.482 | 0.486 | 0.907 | 0.747 | 1.201 | 0.845 |
| | 720 | 0.425 | 0.445 | 0.427 | 0.445 | 0.424 | 0.444 | 0.831 | 0.657 | 0.462 | 0.468 | 0.463 | 0.474 | 0.500 | 0.497 | 0.863 | 0.672 | 0.515 | 0.511 | 0.963 | 0.783 | 3.625 | 1.451 |
| | Avg | 0.371 | 0.400 | 0.383 | 0.407 | 0.386 | 0.411 | 0.509 | 0.515 | 0.414 | 0.427 | 0.437 | 0.449 | 0.439 | 0.452 | 0.602 | 0.543 | 0.450 | 0.459 | 0.826 | 0.703 | 0.749 | 0.623 |
| | Me | 0.386 | 0.409 | 0.404 | 0.416 | 0.464 | 0.423 | 0.536 | 0.509 | 0.427 | 0.433 | 0.446 | 0.457 | 0.458 | 0.459 | 0.573 | 0.532 | 0.469 | 0.469 | 0.848 | 0.715 | 0.775 | 0.696 |
| ETTm1 | 96 | 0.313 | 0.348 | 0.324 | 0.356 | 0.336 | 0.369 | 0.345 | 0.372 | 0.338 | 0.375 | 0.379 | 0.419 | 0.375 | 0.398 | 0.374 | 0.409 | 0.505 | 0.475 | 0.543 | 0.510 | 0.672 | 0.571 |
| | 192 | 0.361 | 0.376 | 0.366 | 0.383 | 0.374 | 0.387 | 0.380 | 0.389 | 0.374 | 0.387 | 0.426 | 0.441 | 0.408 | 0.410 | 0.400 | 0.407 | 0.553 | 0.496 | 0.557 | 0.537 | 0.795 | 0.669 |
| | 336 | 0.383 | 0.396 | 0.395 | 0.403 | 0.408 | 0.407 | 0.413 | 0.413 | 0.410 | 0.411 | 0.445 | 0.459 | 0.435 | 0.428 | 0.438 | 0.438 | 0.621 | 0.537 | 0.754 | 0.655 | 1.212 | 0.871 |
| | 720 | 0.451 | 0.435 | 0.467 | 0.440 | 0.477 | 0.442 | 0.474 | 0.453 | 0.478 | 0.450 | 0.543 | 0.490 | 0.499 | 0.462 | 0.527 | 0.502 | 0.671 | 0.561 | 0.908 | 0.724 | 1.166 | 0.823 |
| | Avg | 0.377 | 0.388 | 0.388 | 0.395 | 0.399 | 0.401 | 0.403 | 0.407 | 0.400 | 0.406 | 0.448 | 0.452 | 0.429 | 0.425 | 0.435 | 0.437 | 0.588 | 0.517 | 0.691 | 0.607 | 0.961 | 0.734 |
| | Me | 0.372 | 0.386 | 0.380 | 0.383 | 0.391 | 0.397 | 0.397 | 0.401 | 0.392 | 0.399 | 0.436 | 0.450 | 0.422 | 0.419 | 0.419 | 0.423 | 0.587 | 0.517 | 0.656 | 0.596 | 0.981 | 0.746 |
| ETTm2 | 96 | 0.174 | 0.257 | 0.180 | 0.262 | 0.184 | 0.267 | 0.193 | 0.292 | 0.187 | 0.267 | 0.203 | 0.287 | 0.189 | 0.280 | 0.209 | 0.308 | 0.255 | 0.339 | 0.435 | 0.507 | 0.365 | 0.453 |
| | 192 | 0.238 | 0.302 | 0.246 | 0.306 | 0.252 | 0.307 | 0.284 | 0.362 | 0.249 | 0.309 | 0.269 | 0.328 | 0.253 | 0.319 | 0.311 | 0.382 | 0.281 | 0.340 | 0.730 | 0.673 | 0.533 | 0.563 |
| | 336 | 0.297 | 0.338 | 0.307 | 0.340 | 0.314 | 0.345 | 0.369 | 0.427 | 0.321 | 0.351 | 0.325 | 0.366 | 0.314 | 0.357 | 0.442 | 0.446 | 0.339 | 0.372 | 1.201 | 0.845 | 1.363 | 0.887 |
| | 720 | 0.390 | 0.393 | 0.408 | 0.403 | 0.412 | 0.402 | 0.554 | 0.522 | 0.408 | 0.403 | 0.421 | 0.415 | 0.414 | 0.413 | 0.675 | 0.587 | 0.433 | 0.432 | 3.625 | 1.451 | 3.379 | 1.338 |
| | Avg | 0.274 | 0.322 | 0.285 | 0.327 | 0.291 | 0.330 | 0.350 | 0.401 | 0.291 | 0.333 | 0.305 | 0.349 | 0.293 | 0.342 | 0.409 | 0.436 | 0.327 | 0.371 | 1.498 | 0.869 | 1.410 | 0.810 |
| | Me | 0.267 | 0.320 | 0.276 | 0.323 | 0.283 | 0.326 | 0.327 | 0.395 | 0.285 | 0.330 | 0.297 | 0.347 | 0.284 | 0.338 | 0.377 | 0.424 | 0.310 | 0.356 | 0.966 | 0.759 | 0.948 | 0.725 |
| Exchange | 96 | 0.080 | 0.205 | 0.084 | 0.207 | 0.086 | 0.206 | 0.088 | 0.218 | 0.107 | 0.234 | 0.148 | 0.278 | 0.085 | 0.204 | 0.116 | 0.262 | 0.197 | 0.323 | 1.748 | 1.105 | 0.847 | 0.752 |
| | 192 | 0.173 | 0.297 | 0.188 | 0.319 | 0.176 | 0.299 | 0.176 | 0.315 | 0.226 | 0.334 | 0.271 | 0.380 | 0.182 | 0.303 | 0.215 | 0.359 | 0.300 | 0.369 | 1.874 | 1.151 | 1.204 | 0.895 |
| | 336 | 0.317 | 0.407 | 0.329 | 0.425 | 0.330 | 0.416 | 0.313 | 0.427 | 0.367 | 0.448 | 0.460 | 0.500 | 0.348 | 0.428 | 0.377 | 0.466 | 0.509 | 0.524 | 1.943 | 1.172 | 1.672 | 1.036 |
| | 720 | 0.779 | 0.670 | 0.779 | 0.670 | 0.828 | 0.689 | 0.839 | 0.695 | 0.964 | 0.746 | 1.195 | 0.841 | 1.025 | 0.774 | 0.831 | 0.699 | 1.447 | 0.941 | 2.085 | 1.206 | 2.478 | 1.310 |
| | Avg | 0.337 | 0.394 | 0.345 | 0.405 | 0.355 | 0.403 | 0.354 | 0.414 | 0.416 | 0.443 | 0.519 | 0.500 | 0.410 | 0.427 | 0.385 | 0.447 | 0.613 | 0.539 | 1.913 | 1.159 | 1.550 | 0.998 |
| | Me | 0.245 | 0.352 | 0.258 | 0.372 | 0.253 | 0.358 | 0.245 | 0.371 | 0.297 | 0.396 | 0.366 | 0.440 | 0.265 | 0.366 | 0.296 | 0.413 | 0.405 | 0.447 | 1.909 | 1.162 | 1.438 | 0.966 |

Table 8: The complete results for LTSF. The results of 4 different prediction lengths of different models are listed in the table. The look-back window sizes are set to 96 for all datasets. We also calculate the average (Avg) and median(Me) of the results for the 4 prediction lengths and the number of optimal values obtained by different models.

| Models | | PDD | | FITS 2024 | | WITRAN 2023 | | DLinear 2023 | | TimesNet 2022 | | FEDformer 2022b | | ETSformer 2022 | | LightTS 2022 | | Autoformer 2021 | | Pyraformer 2021 | | Informer 2022a | |
|---|---|---|---|---|---|---|---|---|---|---|---|---|---|---|---|---|---|---|---|---|---|---|---|
| Metric | | MSE | MAE | MSE | MAE | MSE | MAE | MSE | MAE | MSE | MAE | MSE | MAE | MSE | MAE | MSE | MAE | MSE | MAE | MSE | MAE | MSE | MAE |
| Electricity | 96 | 0.129 | 0.225 | 0.293 | 0.401 | 0.237 | 0.335 | 0.197 | 0.282 | 0.168 | 0.272 | 0.193 | 0.308 | 0.187 | 0.304 | 0.207 | 0.307 | 0.201 | 0.317 | 0.386 | 0.449 | 0.274 | 0.368 |
| | 192 | 0.146 | 0.242 | 0.268 | 0.378 | 0.258 | 0.350 | 0.196 | 0.285 | 0.184 | 0.289 | 0.201 | 0.315 | 0.199 | 0.315 | 0.213 | 0.316 | 0.222 | 0.334 | 0.378 | 0.443 | 0.296 | 0.386 |
| | 336 | 0.154 | 0.253 | 0.355 | 0.452 | 0.273 | 0.362 | 0.209 | 0.301 | 0.198 | 0.300 | 0.214 | 0.329 | 0.212 | 0.329 | 0.230 | 0.333 | 0.231 | 0.338 | 0.376 | 0.443 | 0.300 | 0.394 |
| | 720 | 0.186 | 0.288 | 0.416 | 0.498 | 0.300 | 0.382 | 0.245 | 0.333 | 0.220 | 0.320 | 0.246 | 0.355 | 0.233 | 0.245 | 0.265 | 0.360 | 0.254 | 0.361 | 0.376 | 0.445 | 0.373 | 0.439 |
| | Avg | 0.154 | 0.252 | 0.333 | 0.432 | 0.267 | 0.357 | 0.212 | 0.300 | 0.192 | 0.295 | 0.214 | 0.327 | 0.208 | 0.323 | 0.229 | 0.329 | 0.227 | 0.338 | 0.379 | 0.445 | 0.311 | 0.397 |
| | Me | 0.158 | 0.257 | 0.324 | 0.427 | 0.265 | 0.356 | 0.203 | 0.293 | 0.191 | 0.295 | 0.208 | 0.322 | 0.206 | 0.322 | 0.222 | 0.325 | 0.227 | 0.336 | 0.377 | 0.444 | 0.298 | 0.390 |
| Traffic | 96 | 0.389 | 0.251 | 0.898 | 0.572 | 1.037 | 0.441 | 0.650 | 0.396 | 0.593 | 0.321 | 0.587 | 0.366 | 0.607 | 0.392 | 0.615 | 0.391 | 0.613 | 0.388 | 0.867 | 0.468 | 0.719 | 0.391 |
| | 192 | 0.414 | 0.261 | 0.763 | 0.522 | 1.061 | 0.455 | 0.598 | 0.370 | 0.617 | 0.336 | 0.604 | 0.373 | 0.621 | 0.399 | 0.601 | 0.382 | 0.616 | 0.382 | 0.869 | 0.467 | 0.696 | 0.379 |
| | 336 | 0.429 | 0.267 | 0.894 | 0.608 | 1.095 | 0.470 | 0.605 | 0.373 | 0.629 | 0.336 | 0.621 | 0.383 | 0.622 | 0.399 | 0.613 | 0.386 | 0.622 | 0.337 | 0.881 | 0.469 | 0.777 | 0.420 |
| | 720 | 0.467 | 0.285 | 1.019 | 0.646 | 1.121 | 0.474 | 0.645 | 0.394 | 0.640 | 0.350 | 0.626 | 0.382 | 0.632 | 0.396 | 0.658 | 0.407 | 0.660 | 0.408 | 0.896 | 0.473 | 0.864 | 0.472 |
| | Avg | 0.425 | 0.266 | 0.894 | 0.587 | 1.079 | 0.460 | 0.625 | 0.383 | 0.620 | 0.336 | 0.610 | 0.376 | 0.621 | 0.396 | 0.622 | 0.392 | 0.628 | 0.379 | 0.878 | 0.469 | 0.764 | 0.416 |
| | Me | 0.432 | 0.265 | 0.879 | 0.597 | 1.078 | 0.463 | 0.625 | 0.384 | 0.623 | 0.336 | 0.613 | 0.378 | 0.622 | 0.396 | 0.614 | 0.389 | 0.619 | 0.385 | 0.875 | 0.469 | 0.748 | 0.406 |
| Weather | 96 | 0.151 | 0.199 | 0.174 | 0.214 | 0.178 | 0.223 | 0.196 | 0.255 | 0.172 | 0.220 | 0.217 | 0.296 | 0.197 | 0.281 | 0.182 | 0.242 | 0.266 | 0.336 | 0.622 | 0.556 | 0.300 | 0.384 |
| | 192 | 0.195 | 0.246 | 0.221 | 0.254 | 0.223 | 0.261 | 0.237 | 0.296 | 0.219 | 0.261 | 0.276 | 0.336 | 0.237 | 0.312 | 0.227 | 0.287 | 0.307 | 0.367 | 0.739 | 0.624 | 0.598 | 0.544 |
| | 336 | 0.242 | 0.289 | 0.278 | 0.309 | 0.288 | 0.309 | 0.283 | 0.335 | 0.280 | 0.306 | 0.339 | 0.380 | 0.298 | 0.353 | 0.282 | 0.334 | 0.359 | 0.395 | 1.004 | 0.753 | 0.578 | 0.523 |
| | 720 | 0.315 | 0.344 | 0.358 | 0.349 | 0.372 | 0.363 | 0.345 | 0.381 | 0.365 | 0.359 | 0.403 | 0.428 | 0.352 | 0.390 | 0.352 | 0.386 | 0.419 | 0.428 | 1.420 | 0.934 | 1.059 | 0.741 |
| | Avg | 0.226 | 0.270 | 0.258 | 0.278 | 0.265 | 0.289 | 0.265 | 0.317 | 0.259 | 0.287 | 0.309 | 0.360 | 0.271 | 0.334 | 0.261 | 0.312 | 0.338 | 0.382 | 0.946 | 0.717 | 0.634 | 0.548 |
| | Me | 0.230 | 0.277 | 0.250 | 0.275 | 0.255 | 0.285 | 0.260 | 0.316 | 0.250 | 0.284 | 0.308 | 0.358 | 0.268 | 0.333 | 0.255 | 0.311 | 0.333 | 0.381 | 0.872 | 0.689 | 0.588 | 0.534 |
| ETTh1 | 96 | 0.371 | 0.391 | 0.381 | 0.391 | 0.414 | 0.419 | 0.386 | 0.400 | 0.384 | 0.402 | 0.376 | 0.419 | 0.494 | 0.479 | 0.424 | 0.432 | 0.449 | 0.459 | 0.664 | 0.612 | 0.865 | 0.713 |
| | 192 | 0.427 | 0.425 | 0.443 | 0.422 | 0.464 | 0.448 | 0.437 | 0.432 | 0.436 | 0.429 | 0.420 | 0.439 | 0.538 | 0.504 | 0.475 | 0.462 | 0.500 | 0.482 | 0.790 | 0.681 | 1.008 | 0.792 |
| | 336 | 0.465 | 0.441 | 0.474 | 0.446 | 0.516 | 0.478 | 0.481 | 0.459 | 0.477 | 0.456 | 0.459 | 0.465 | 0.574 | 0.521 | 0.518 | 0.488 | 0.521 | 0.496 | 0.891 | 0.738 | 1.107 | 0.809 |
| | 720 | 0.471 | 0.466 | 0.464 | 0.463 | 0.538 | 0.509 | 0.519 | 0.516 | 0.521 | 0.500 | 0.459 | 0.474 | 0.562 | 0.535 | 0.547 | 0.533 | 0.514 | 0.512 | 0.963 | 0.782 | 1.181 | 0.865 |
| | Avg | 0.433 | 0.430 | 0.438 | 0.431 | 0.483 | 0.464 | 0.456 | 0.452 | 0.444 | 0.447 | 0.429 | 0.449 | 0.542 | 0.510 | 0.491 | 0.479 | 0.496 | 0.487 | 0.827 | 0.703 | 1.040 | 0.795 |
| | Me | 0.458 | 0.441 | 0.459 | 0.434 | 0.490 | 0.463 | 0.459 | 0.446 | 0.456 | 0.445 | 0.440 | 0.452 | 0.550 | 0.513 | 0.497 | 0.475 | 0.507 | 0.489 | 0.841 | 0.710 | 1.058 | 0.801 |
| ETTh2 | 96 | 0.287 | 0.339 | 0.290 | 0.339 | 0.325 | 0.364 | 0.333 | 0.387 | 0.340 | 0.374 | 0.358 | 0.397 | 0.340 | 0.391 | 0.397 | 0.437 | 0.346 | 0.388 | 0.645 | 0.597 | 3.755 | 1.525 |
| | 192 | 0.364 | 0.391 | 0.375 | 0.388 | 0.433 | 0.427 | 0.477 | 0.476 | 0.402 | 0.414 | 0.429 | 0.439 | 0.430 | 0.439 | 0.520 | 0.504 | 0.456 | 0.452 | 0.788 | 0.683 | 5.602 | 1.931 |
| | 336 | 0.409 | 0.427 | 0.414 | 0.425 | 0.471 | 0.457 | 0.594 | 0.541 | 0.452 | 0.452 | 0.496 | 0.487 | 0.485 | 0.479 | 0.626 | 0.559 | 0.482 | 0.486 | 0.907 | 0.747 | 4.721 | 1.835 |
| | 720 | 0.425 | 0.445 | 0.419 | 0.437 | 0.499 | 0.480 | 0.831 | 0.657 | 0.424 | 0.444 | 0.463 | 0.474 | 0.500 | 0.497 | 0.863 | 0.672 | 0.515 | 0.511 | 0.963 | 0.783 | 3.647 | 1.625 |
| | Avg | 0.371 | 0.400 | 0.375 | 0.397 | 0.432 | 0.432 | 0.559 | 0.515 | 0.414 | 0.427 | 0.437 | 0.449 | 0.439 | 0.452 | 0.602 | 0.543 | 0.450 | 0.459 | 0.826 | 0.703 | 4.431 | 1.729 |
| | Me | 0.386 | 0.409 | 0.395 | 0.406 | 0.452 | 0.442 | 0.536 | 0.509 | 0.427 | 0.433 | 0.446 | 0.457 | 0.458 | 0.459 | 0.573 | 0.532 | 0.469 | 0.469 | 0.848 | 0.715 | 4.238 | 1.730 |
| ETTm1 | 96 | 0.313 | 0.348 | 0.351 | 0.370 | 0.375 | 0.402 | 0.345 | 0.372 | 0.338 | 0.375 | 0.379 | 0.419 | 0.375 | 0.398 | 0.374 | 0.409 | 0.505 | 0.475 | 0.543 | 0.510 | 0.672 | 0.571 |
| | 192 | 0.361 | 0.376 | 0.392 | 0.393 | 0.427 | 0.434 | 0.380 | 0.389 | 0.374 | 0.387 | 0.426 | 0.441 | 0.408 | 0.410 | 0.400 | 0.407 | 0.553 | 0.496 | 0.557 | 0.537 | 0.795 | 0.669 |
| | 336 | 0.383 | 0.396 | 0.424 | 0.413 | 0.455 | 0.452 | 0.413 | 0.413 | 0.408 | 0.407 | 0.445 | 0.459 | 0.435 | 0.428 | 0.438 | 0.438 | 0.621 | 0.537 | 0.754 | 0.655 | 1.212 | 0.871 |
| | 720 | 0.451 | 0.435 | 0.485 | 0.448 | 0.527 | 0.488 | 0.474 | 0.453 | 0.478 | 0.442 | 0.543 | 0.490 | 0.499 | 0.462 | 0.527 | 0.502 | 0.671 | 0.561 | 0.908 | 0.724 | 1.166 | 0.823 |
| | Avg | 0.377 | 0.388 | 0.413 | 0.406 | 0.446 | 0.444 | 0.403 | 0.407 | 0.400 | 0.403 | 0.448 | 0.452 | 0.429 | 0.425 | 0.435 | 0.439 | 0.588 | 0.517 | 0.691 | 0.607 | 0.961 | 0.734 |
| | Me | 0.380 | 0.393 | 0.408 | 0.403 | 0.441 | 0.443 | 0.397 | 0.401 | 0.391 | 0.397 | 0.436 | 0.450 | 0.422 | 0.419 | 0.419 | 0.423 | 0.587 | 0.517 | 0.656 | 0.596 | 0.981 | 0.746 |
| ETTm2 | 96 | 0.174 | 0.257 | 0.181 | 0.264 | 0.191 | 0.272 | 0.193 | 0.292 | 0.187 | 0.267 | 0.203 | 0.287 | 0.189 | 0.280 | 0.209 | 0.308 | 0.255 | 0.339 | 0.435 | 0.507 | 0.365 | 0.453 |
| | 192 | 0.238 | 0.302 | 0.246 | 0.304 | 0.261 | 0.316 | 0.284 | 0.362 | 0.249 | 0.307 | 0.269 | 0.328 | 0.253 | 0.319 | 0.311 | 0.382 | 0.281 | 0.340 | 0.730 | 0.673 | 0.533 | 0.563 |
| | 336 | 0.297 | 0.338 | 0.306 | 0.341 | 0.330 | 0.358 | 0.369 | 0.427 | 0.321 | 0.351 | 0.325 | 0.366 | 0.314 | 0.357 | 0.442 | 0.446 | 0.339 | 0.372 | 1.201 | 0.845 | 1.363 | 0.887 |
| | 720 | 0.390 | 0.393 | 0.407 | 0.397 | 0.450 | 0.427 | 0.554 | 0.522 | 0.408 | 0.403 | 0.421 | 0.415 | 0.414 | 0.413 | 0.675 | 0.587 | 0.433 | 0.432 | 3.625 | 1.451 | 3.379 | 1.338 |
| | Avg | 0.274 | 0.322 | 0.285 | 0.327 | 0.308 | 0.343 | 0.350 | 0.401 | 0.291 | 0.333 | 0.305 | 0.349 | 0.293 | 0.342 | 0.409 | 0.436 | 0.327 | 0.371 | 1.498 | 0.869 | 1.410 | 0.810 |
| | Me | 0.276 | 0.323 | 0.276 | 0.323 | 0.296 | 0.337 | 0.327 | 0.395 | 0.285 | 0.330 | 0.297 | 0.347 | 0.284 | 0.338 | 0.377 | 0.424 | 0.310 | 0.356 | 0.966 | 0.759 | 0.948 | 0.725 |

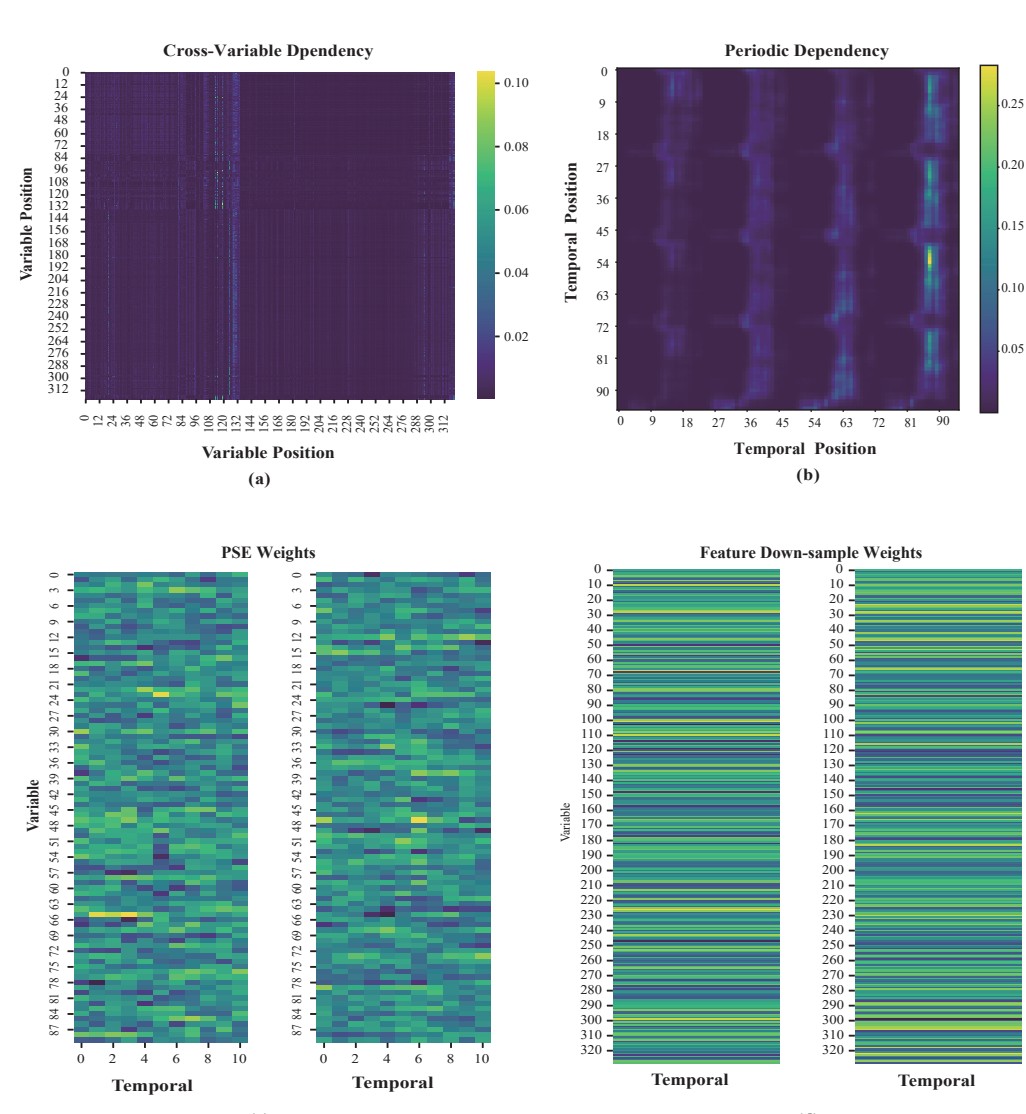

Figure 7: Visualisation of dependencies captured by PDD. (a) Attention-based cross-variable dependency. (b) Attention-based periodic dependency. (c) Convolutional kernel weights within the PSE. (d) Feature down-sampling (FDS) weights.

## E.1 RESISTANCE TO GAUSSIAN NOISE

We inject Gaussian noise with $\sigma \in \{0.0, 0.1, \ldots, 1.0\}$ into the Electricity dataset via $x'_t = x_t + \epsilon$, $\epsilon \sim \mathcal{N}(0, \sigma^2)$. Even at $\sigma = 1.0$, performance deteriorates only marginally, indicating graceful degradation across prediction lengths.

## E.2 TOLERANCE TO MISSING VALUES

We randomly mask a proportion $m \in \{0.0, 0.1, 0.3, 0.5, 0.7\}$ of each input sequence. PDD remains resilient even with 70% missing values, thanks to its residual refinement design that first captures stable permutation-invariant structures.

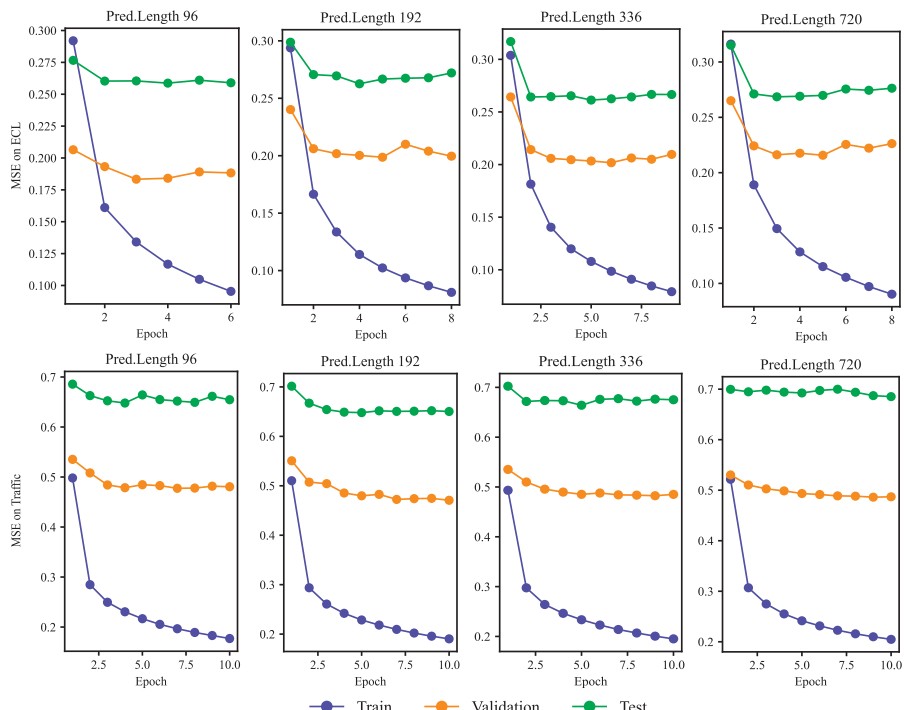

Figure 8: Training trajectories of the full Transformer on Electricity and Traffic datasets. Training stops once early-stopping patience (three epochs) is exceeded.

## F  TRANSFORMER LIMITATIONS ANALYSIS

Transformer-based methods dominate multivariate time-series forecasting yet can struggle to exploit historical context due to gradient conflicts arising from mixed dependency modelling (Zeng et al., 2023; Liu et al., 2024; Gao et al., 2025). We compare the full encoder–decoder Transformer with a decoder-only variant that receives minimal historical information.

Figure 8 shows training trajectories on the Electricity and Traffic datasets under a learning rate of $1 \times 10^{-4}$. The encoder– decoder configuration rapidly overfits, as evidenced by a steep decline in training loss but stagnant validation loss.

Figure 9 contrasts the forecasting accuracy of the two models. The full Transformer occasionally outperforms the decoder-only version yet the gap is narrow and sometimes reversed, indicating inefficient use of historical signals. These findings align with prior observations that simply extending context length yields limited gains without careful dependency decoupling.

## G  SUMMARY OF VISUAL ANALYSIS

The qualitative forecasts in Figures 10–12 illustrate that PDD tracks complex dynamics across datasets. On `Traffic`, PDD captures sharp transitions and periodic structure. On `Electricity`, predictions align closely with ground truth except for a few abrupt spikes (highlighted in red). On `Weather`, despite high volatility, PDD still follows primary trends, demonstrating resilience to irregular dynamics.

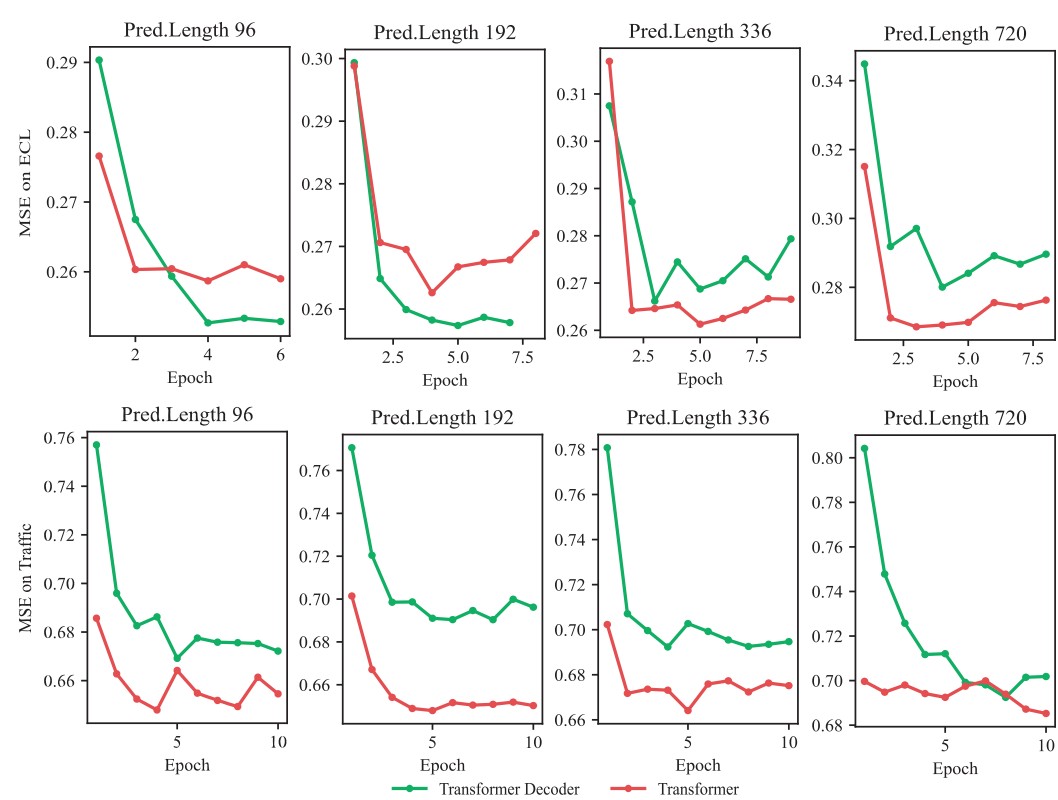

Figure 9: Comparison between encoder–decoder and decoder-only Transformers on the Electricity (top) and Traffic (bottom) datasets.

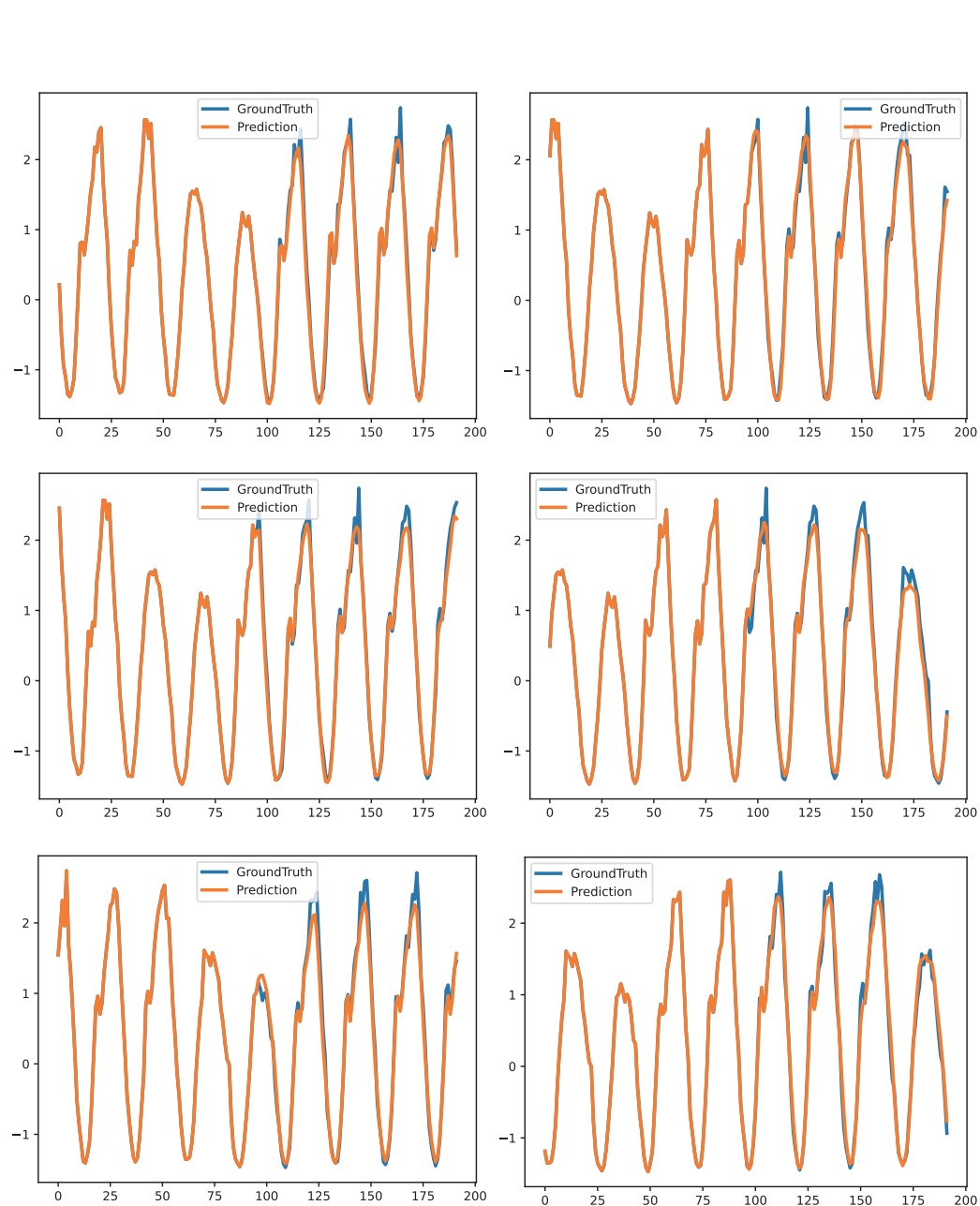

Figure 10: Forecast visualisation on the Traffic dataset.

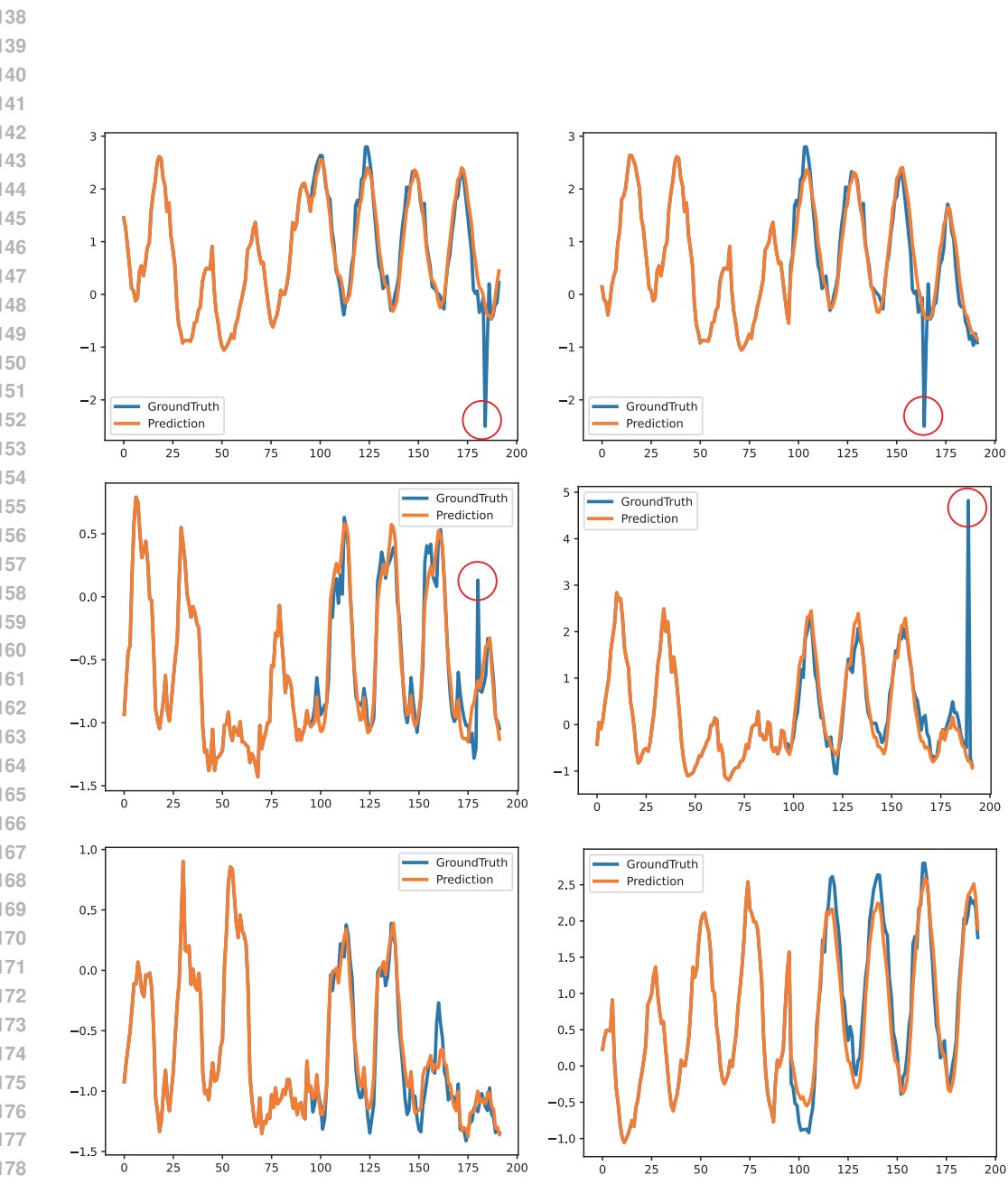

Figure 11: Forecast visualisation on the Electricity dataset. Red circles mark isolated discrepancies.

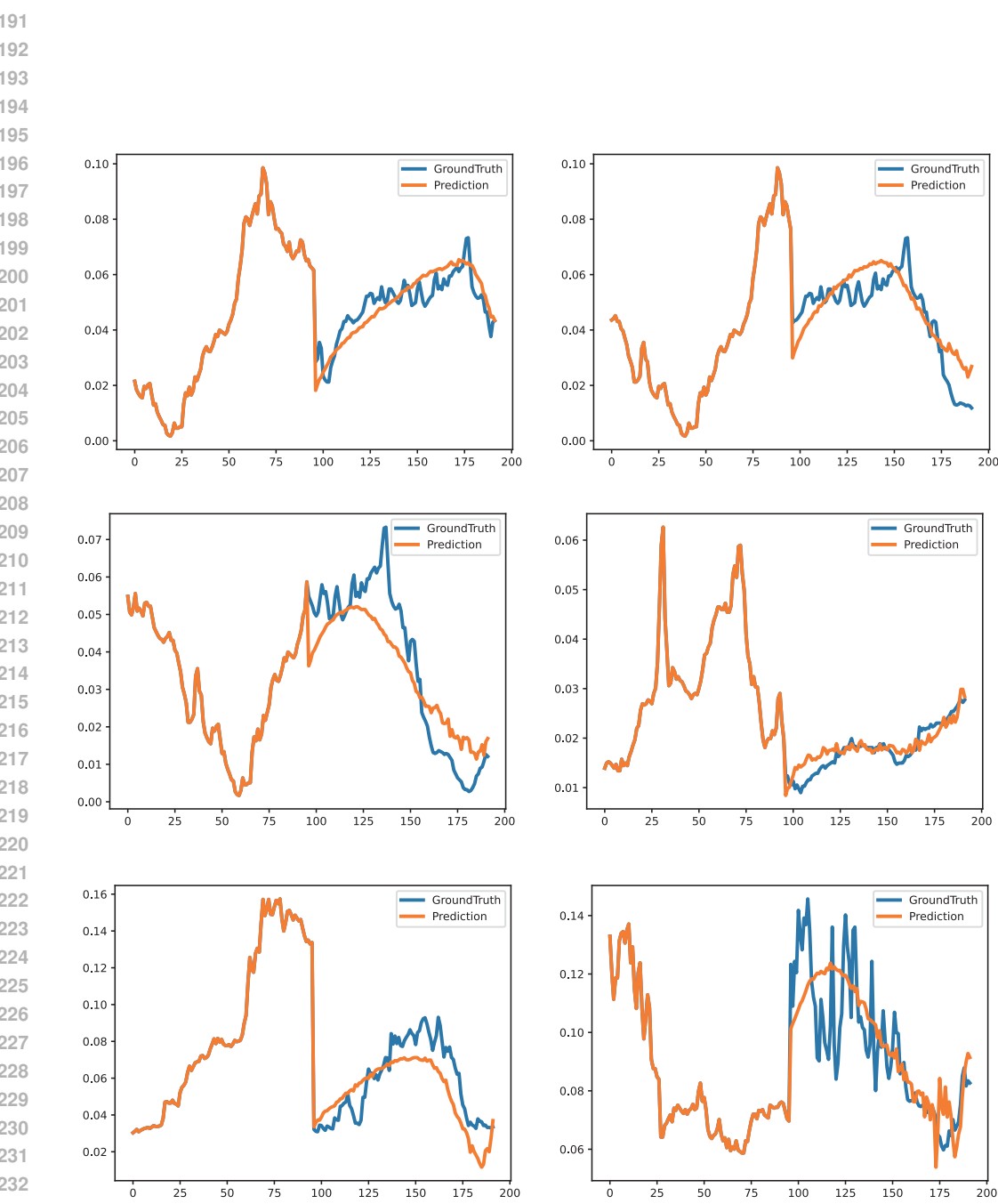

Figure 12: Forecast visualisation on the Weather dataset.