# OpenReview forum: "Decoupling Permutation-Invariant and Permutation-Sensitive Dependencies for Time-Series Forecasting"
_ICLR.cc/2026/Conference — ICLR 2026 Conference Withdrawn Submission_

### Official Review · Reviewer_V4nH · 2025-10-29

**Soundness:** 2
**Presentation:** 1
**Contribution:** 2
**Rating:** 2
**Confidence:** 3

**Summary:**

The authors claim that multivariate time series forecasting suffers from gradient conflicts when jointly modeling permutation-invariant dependencies (PID) and permutation-sensitive dependencies (PSD). They propose Permutation Dependency Decoupling (PDD), a two-stage framework that first learns stable, order-agnostic patterns through a Permutation-Invariant Encoder (PIE), then refines predictions by capturing temporal dynamics via a Permutation-Sensitive Encoder (PSE) while freezing PIE parameters. The authors introduce the Temporal Order Sensitivity Test (TOST), which applies consistent permutations during both training and inference to distinguish genuine temporal modeling from memorization. Experiments across eight benchmarks demonstrate state-of-the-art performance and plug-and-play extensibility for existing models. The authors argue their method addresses fundamental optimization conflicts in Transformer-based forecasting without sacrificing either dependency type.

**Strengths:**

This paper provides empirical evidence of gradient conflicts between permutation-invariant and permutation-sensitive dependencies, effectively motivating the decoupling approach—though theoretical justification for why two-stage training resolves these conflicts remains absent. Proposed TOST evaluation methodology addresses limitations in prior temporal sensitivity tests by applying consistent permutations during both training and inference, revealing that many existing models primarily learn order-agnostic patterns. The paper demonstrates various empirical results across multiple benchmarks with practical plug-and-play extensibility that substantially improves existing models.

**Weaknesses:**

The proposed method is interesting and the performance improvements are impressive, but the current manuscript suffers from several fatal issues that fall short of publication quality.

### W1: Critical Experimental Flaws.
Time series forecasting is sensitive to initial weights, making multiple seed experiments essential, yet no such information is provided. Neither p-values for performance differences nor standard deviations from multiple runs are reported, which is a serious omission for reproducibility. Most baselines are from before 2023, while TSF models have significantly advanced since 2024—recent strong baselines like TimeMixer++ (ICLR 2025) must be included. Given the focus on decomposition, comparisons with classical decomposition methods and decomposition-based TSF approaches are also necessary. The paper lacks clarity on whether baseline results were obtained through direct experiments with identical hyperparameters or taken from other publications. Training details such as the number of epochs must be explicitly specified in the Appendix for reproducibility. Additionally, the median values in Tables 7 and 8 are inconsistent and require justification. Including commonly-used Solar and PEMS datasets in the benchmark is strongly recommended.

### W2: Weak Theoretical Foundation and Unclear PID/PSD Separation.
The claim that stationary patterns correspond to PID and non-stationary patterns to PSD lacks explanation, especially when typical TSF papers aim to reduce non-stationarity through stationarization. The assertion that PID characterizes "stable statistical properties" is superficial without proper justification (line 195). Terms like "fixed structure" and "evolving dependency" are poorly defined without explanation (line 13). Fundamentally, many patterns exhibit both characteristics simultaneously, making the PID/PSD separation ambiguous—it remains unclear which dependencies are learned by which encoder. There is no theoretical analysis of why two-stage training resolves gradient conflicts or mathematical proof that freezing PIE is optimal. Comparisons with multi-task learning gradient surgery techniques (e.g., PCGrad, GradNorm, ConFIG: Conflict-Free Training of PINN) are necessary.

### W3: Overclaiming Efficiency and Robustness.
Claiming robustness based solely on PDD's graceful degradation under noise and missing data, without comparison to other models under identical conditions, constitutes significant overclaiming. No quantitative comparisons of training costs—including model parameters, training time, computational complexity, and peak memory—are provided. Given PDD's two-stage training, detailed and concrete training cost analysis is essential.

Additional issues include notation conflicts between lines 154 and 198, poor definition of periodic dependency at line 159, pervasive inappropriate citations throughout the paper suggesting lack of review, and unclear inputs for HSTD and PSTD in Figure 3. Overall writing quality is insufficient and requires thorough revision.

**Questions:**

See weakness

---

### Official Review · Reviewer_DY5e · 2025-10-31

**Soundness:** 3
**Presentation:** 3
**Contribution:** 3
**Rating:** 4
**Confidence:** 3

**Summary:**

The paper argues that mixing permutation-invariant (PID: stable, order-agnostic patterns) and permutation-sensitive (PSD: order-dependent dynamics) within a single forecaster induces gradient conflicts, harming optimization. It proposes Permutation Dependency Decoupling (PDD): (i) a Permutation-Invariant Encoder (PIE) trained first and then frozen, and (ii) a Permutation-Sensitive Encoder (PSE) that refines the PIE forecast via a history–prediction fusion (“correction”) pathway. A Temporal Order Sensitivity Test (TOST) permutes timestamps consistently at train and test to probe whether a model genuinely leverages temporal order. Empirically, PDD reports top results on several benchmarks and improves several strong baselines when added as a plug-in.

**Strengths:**

- Clear decoupling objective + optimization story. The paper formalizes PID vs. PSD, then maps this to a staged training procedure (PIE to freeze to PSE), with evidence of negative gradient cosine similarity when trained jointly (Fig. 1). The staged design is technically simple and easy to adopt.
- Architectural choices are pragmatic. PIE uses three “perspectives” (time-wise, channel-wise, head-wise) with negligible extra parameters (<0.001%) via shared weights and routing; PSE reframes forecasting as an iterative correction over joint history–prediction context.
- TOST is a useful diagnostic. Training/testing under the same permutation avoids the usual “permute-only-at-inference” pitfall and gives a more controlled measure of temporal-order reliance.
- PDD is competitive or best on many horizons/datasets; adding PDD to iTransformer/DLinear/Client/PatchTST improves MSE on ECL.

**Weaknesses:**

- Definition/treatment of “permutation-invariance” may conflate effects. PID includes cross-variable correlations and periodicities but is modeled by removing temporal order and aggregating via attention from multiple perspectives. Some periodic/seasonal structure is time-relative; treating it as strictly order-agnostic could undercut the conceptual neatness (e.g., phase matters). A short analysis clarifying which periodicities survive order removal would help.
- TOST still has edge cases. TOST permutes consistently at train/test, which is good; however, some datasets (e.g., Exchange) may contain regime shifts or calendar effects where fixed permutations alter distributional alignment in non-temporal ways (weekday/weekend, holidays). A note on which datasets TOST is most reliable for, and sensitivity to multiple independent permutations, would calibrate interpretation.
- It would be nice to see some comparisons with some more recent models as well
- Ablation granularity. We see decoupled vs. joint, order (PIE to PSE vs. PSE to PIE), and hard results; less clear is the separate contribution of (i) the three-expert PIE routing, (ii) freezing vs. partial unfreezing, and (iii) each PSE component (HSTD vs. PSTD vs. FDS). A finer ablation would isolate where the gains come from.

**Questions:**

- PSE is framed as an iterative correction. If predictions are “refined” across layers/iterations, what is the retry/rollback policy when corrections overshoot or destabilize?
- Complexity & scaling. Can you provide time/space complexity in terms of L (history), D (variables), and O (horizon) for PIE and PSE separately?
- For Table 1 and the full results, were training/inference budgets matched (epochs, early-stopping patience, FLOPs/forward passes) across baselines?
- PSE is described as an iterative correction. Do you perform any on-the-fly re-correction or rollback when corrections degrade loss? If yes, what’s the iteration budget and wall-clock impact? If not, please clarify why a single pass suffices.

---

### Official Review · Reviewer_HjNb · 2025-11-01

**Soundness:** 2
**Presentation:** 3
**Contribution:** 2
**Rating:** 2
**Confidence:** 5

**Summary:**

The paper approaches the problem from the perspective of gradient conflict, proposing the Permutation Dependency Decoupling (PDD) method and Temporal Order Sensitivity Test (TOST), and attempts to validate the effectiveness of the proposed methods through experiments.

**Strengths:**

S1: The motivation of the paper is interesting.

S2: The author's description is relatively clear.

**Weaknesses:**

W1: The author’s review of past methods is insufficient and lacks important references. For example, methods based on TCN: ModernTCN[1], methods based on RNN: PGN[2], methods based on Linear structures such as CycleNet[3] and TimeKAN[4], and methods based on attention mechanisms such as Leddam[5].

[1] ModernTCN: A modern pure convolution structure for general time series analysis. In The Twelfth International Conference on Learning Representations.

[2] PGN: The RNN's New Successor is Effective for Long-Range Time Series Forecasting. In The Thirty-eighth Annual Conference on Neural Information Processing Systems.

[3] CycleNet: Enhancing Time Series Forecasting through Modeling Periodic Patterns. In The Thirty-eighth Annual Conference on Neural Information Processing Systems.

[4] TimeKAN: KAN-based Frequency Decomposition Learning Architecture for Long-term Time Series Forecasting. In The Thirteenth International Conference on Learning Representations.

[5] Revitalizing Multivariate Time Series Forecasting: Learnable Decomposition with Inter-series Dependencies and Intra-series Variations Modeling. In Proceedings of the 41st International Conference on Machine Learning.

W2: Based on W1, the experiment also lacks comparisons with these important baseline methods.

W3: The experimental section of the paper contains significant flaws. The authors claim that their hyperparameters were selected from a predefined candidate set based on the validation set loss performance. However, the table provided by the authors does not show a sufficiently detailed search space, such as the d_models, e_layers, etc., involved in the models. Moreover, it is unclear what the best hyperparameters for the selected baseline comparisons are. If the authors indeed performed hyperparameter search on the validation set, they should provide the complete search space and the final selected parameters for each task across different datasets. If no such search was conducted, the experimental results are not reliable. Even with identical parameters and random seeds, results can vary across different hardware platforms. To eliminate the influence of platform and hyperparameter sensitivity on model performance, and to ensure a fair comparison of all models, all methods should be reproduced on the author’s same platform, sharing a sufficiently broad hyperparameter search space. The best parameters should be selected on the validation set and then used for evaluation on the test set.

W4: The credibility of the results concerning model efficiency is low. To compare model efficiency, the impact of hyperparameter size itself must be eliminated, which differs from the validation method for experimental results. Otherwise, by specifically choosing smaller parameters for certain tasks, one could claim high efficiency, which is extremely unrigorous and unfair.

**Questions:**

The authors need to provide more experimental results to fairly validate the performance and efficiency of their method.

---

### Official Review · Reviewer_k3Eo · 2025-11-02

**Soundness:** 3
**Presentation:** 3
**Contribution:** 3
**Rating:** 4
**Confidence:** 3

**Summary:**

This paper tackles a critical problem in multivariate time-series forecasting: the entanglement between permutation-invariant dependencies (PID) and permutation-sensitive dependencies. The authors argue that jointly optimizing both leads to gradient conflicts that degrade generalization. To address this, they propose Permutation Dependency Decoupling (PDD), a gradient-level training framework that explicitly separates invariant and order-sensitive components. Experiments on eight standard benchmarks (ECL, Traffic, Weather, Exchange, ETT variants) show consistent improvements over leading models (Client, iTransformer, TimesNet, FEDformer, PatchTST, etc.), along with strong efficiency and robustness.

**Strengths:**

1. The notion of explicitly decoupling permutation-invariant and permutation-sensitive dependencies is well-motivated. Prior work decomposed by signal attributes (trend, seasonality) or by scale, but not by gradient-level permutation sensitivity.
2. PDD’s plug-and-play extensibility indicates practical value. It improves diverse baselines without retraining them from scratch.
3. Comprehensive experiments across diverse datasets, prediction lengths, and robustness settings (noise, missing data) convincingly demonstrate performance gains.
4. Ablation studies and ordering experiments provide strong empirical justification for the decoupling strategy.

**Weaknesses:**

1. The framework’s intuition is strong, but the paper lacks a formal convergence or conflict-reduction proof.
2. While average performance is excellent, the paper could analyze scenarios where PDD underperforms, for instance, extremely short sequences or purely periodic synthetic data, to clarify boundaries of applicability.
3. The multi-expert routing in PIE is an appealing idea but lacks quantitative ablation. What happens if only time-wise or channel-wise routing is used? Such analysis would clarify how each expert contributes to invariant modeling.

**Questions:**

1. Can the authors visualize or quantify how gradient similarity between PID and PSD evolves during training beyond the single example in Fig. 1? Does the decoupling fully eliminate negative correlation or merely reduce it?

---

### Note · Authors · 2025-11-27

I have read and agree with the venue's withdrawal policy on behalf of myself and my co-authors.